# Amoeboid-like migration ensures correct horizontal cell layer formation in the developing vertebrate retina

Rana Amini[1], Archit Bhatnagar[1], Raimund Schlüßler[2], Stephanie Möllmert[2,3], Jochen Guck[2,3,4], Caren Norden[1,5]*

[1]Max Planck Institute of Molecular Cell Biology and Genetics, Dresden, Germany; [2]Biotechnology Center, Center for Molecular and Cellular Bioengineering, Technische Universität Dresden, Dresden, Germany; [3]Max Planck Institute for the Science of Light and Max-Planck-Zentrum für Physik und Medizin, Erlangen, Germany; [4]Physics of Life, Technische Universität Dresden, Dresden, Germany; [5]Instituto Gulbenkian de Ciência, Rua da Quinta Grande 6, Oeiras, Portugal

**Abstract** Migration of cells in the developing brain is integral for the establishment of neural circuits and function of the central nervous system. While migration modes during which neurons employ predetermined directional guidance of either preexisting neuronal processes or underlying cells have been well explored, less is known about how cells featuring multipolar morphology migrate in the dense environment of the developing brain. To address this, we here investigated multipolar migration of horizontal cells in the zebrafish retina. We found that these cells feature several hallmarks of amoeboid-like migration that enable them to tailor their movements to the spatial constraints of the crowded retina. These hallmarks include cell and nuclear shape changes, as well as persistent rearward polarization of stable F-actin. Interference with the organization of the developing retina by changing nuclear properties or overall tissue architecture hampers efficient horizontal cell migration and layer formation showing that cell-tissue interplay is crucial for this process. In view of the high proportion of multipolar migration phenomena observed in brain development, the here uncovered amoeboid-like migration mode might be conserved in other areas of the developing nervous system.

*For correspondence:
cnorden@igc.gulbenkian.pt

Competing interest: The authors declare that no competing interests exist.

## Editor's evaluation

The authors probe the role of multipolar migration of horizontal cells in the zebrafish retina. The results reveal amoeboid-like migration enabling cell movements to adapt to environmental spatial constraints in the crowded retina including cell and nuclear shape changes and rearward polarization of stable F-actin.

## Introduction

As neuronal precursors and neurons are often born at different positions from where they ultimately function, their precise migration is key to ensure successful nervous system development. A variety of migration modes featuring diverse cell morphologies ranging from unipolar, bipolar, to multipolar have been uncovered (*Marin and Rubenstein, 2003*; *Tabata and Nakajima, 2003*; *Kawaji et al., 2004*; *Ayala et al., 2007*; *Tanaka et al., 2009*; *Cooper, 2013*; *Rahimi-Balaei et al., 2018*; *Gressens, 2000*). As of yet however, most research unveiling the cellular and molecular mechanisms of migration in the nervous system has focused on radial migration which entails movements perpendicular to the

tissue surface (*Angevine and Sidman, 1961*; *Berry and Rogers, 1965*; *Morest, 1970b*; *Morest, 1970a*; *Rakic, 1971*; *Walsh and Cepko, 1988*; *Marin and Rubenstein, 2001*; *Nadarajah et al., 2001*; *Nadarajah and Parnavelas, 2002*; *Cooper, 2013*).

During radial migration, neurons determine their direction of movement by two different strategies: (1) by establishing process(es) that anchor the migrating neurons to the tissue lamina(e), and thereby facilitating their faithful arrival at their destination (somal translocation) (*Nadarajah et al., 2003*), or (2) by moving along radially oriented fibers of neural progenitors known as radial glia that provide physical scaffolding for migrating neurons (glia-guided migration) (*Rakic, 1971*; *Rakic, 1972*; *Edmondson and Hatten, 1987*; *Hatten, 1990*; *Gertz and Kriegstein, 2015*). In both scenarios, the migrating neurons exhibit elongated unipolar or bipolar morphologies in the direction of travel and move unidirectionally via radial paths to their final positions (*Nadarajah et al., 2001*; *Nadarajah et al., 2003*).

In contrast, neurons that display multipolar morphology are neither attached to the tissue lamina(e) nor move along radial glia fibers. Instead, these cells were shown to extend multiple dynamic processes in various directions (*Tabata and Nakajima, 2003*; *Honda et al., 2003*; *Stensaas, 1967*; *Shoukimas and Hinds, 1978*; *Gadisseux et al., 1990*; *Noctor et al., 2004*; *Nowakowski and Rakic, 1979*). Examples include interneurons of the mammalian neocortex (*Nadarajah et al., 2003*; *Tabata and Nakajima, 2003*; *Tanaka et al., 2006*; *Tanaka et al., 2009*) or pyramidal and granule neurons in the mammalian hippocampus (*Kitazawa et al., 2014*; *Namba et al., 2019*). Despite this prevalence, it is not yet fully understood, how multipolar neurons reach their destination with no predisposed migratory information or scaffolds. It is particularly not known whether and how the dynamics and properties of the densely packed environment of the developing brain (*Bondareff and Narotzky, 1972*; *Sekine et al., 2011*) influence multipolar migration mode, path, and/or efficiency. This knowledge gap is partly due to the inaccessibility of many brain regions for in vivo imaging.

One central nervous system (CNS) region that allows for in vivo imaging in a quantitative manner is the zebrafish retina (*Galli-Resta et al., 2008*). The mature retina consists of five major neuronal types: photoreceptor (PR), horizontal cell (HC), bipolar cell (BC), amacrine cell (AC), retinal ganglion cell (RGC), and a single glial cell-type, Müller glia (MG). During development, retinal neurons move from their birth-site to their ultimate functioning locations (*Figure 1A*), and reproducibly assemble into three distinct nuclear layers: outer nuclear layer (ONL), inner nuclear layer (INL), ganglion cell layer (GCL) (*Figure 1A'*). Synapses between these layers form at two nuclei-free layers, known as plexiform layers: the outer plexiform layer (OPL) and the inner plexiform layer (IPL) (*Figure 1A'*).

Retinal neurons follow diverse and complex migratory modes and routes to reach their destinations (*Chow et al., 2015*; *Icha et al., 2016a*, *Amini et al., 2017*; *Amini et al., 2019*; *Rocha-Martins et al., 2021*). However, parameters that ensure successful migration of diverse neuronal cell types in the complex and highly dynamic environment of the retina are only beginning to be understood (*Chow et al., 2015*; *Icha et al., 2016a*, *Amini et al., 2019*).

Especially intriguing is the movement of HCs, the retinal interneurons that modulate information flow from PRs to BCs (*Chaya et al., 2017*). Division of neuronal progenitors generates committed HC precursors (HCprs) at the apical surface of the retina. HCprs initially migrate radially from their birth-site to the center of the INL while keeping an attachment to the apical surface and featuring a bipolar morphology (Phase 1). Upon detachment of this anchorage (*Figure 1B*, *Figure 1—figure supplement 1C*, *Video 2*), HCprs acquire a multipolar morphology (*Hinds and Hinds, 1979*) and migrate with frequent direction changes (Phase 2) (*Chow et al., 2015*; *Amini et al., 2019*). During this phase, HCprs move deeper into the INL before turning apically toward the HC layer at the upper border of the INL (*Figure 1B*, *Figure 1—figure supplement 1A-B*). En route to their destination, HCprs undergo a terminal division (*Edqvist and Hallböök, 2004*; *Weber et al., 2014*; *Chow et al., 2015*; *Amini et al., 2019*) at diverse depths within the INL giving rise to two postmitotic HCs which migrate toward the HC layer (*Godinho et al., 2007*). We previously showed that migration behavior of HCprs and postmitotic HCs is similar (*Amini et al., 2019*), which is why we will use the term HC from here on in this manuscript to refer to both. It is currently not known how these multipolar cells adapt their migration behavior and trajectories to the crowded environment of the retina (*Matejčić et al., 2018*). The fact that HCs follow unpredictable migration paths despite an overall directionality (*Amini et al., 2019*) suggests that their path selection is not intrinsically programmed but that the surrounding

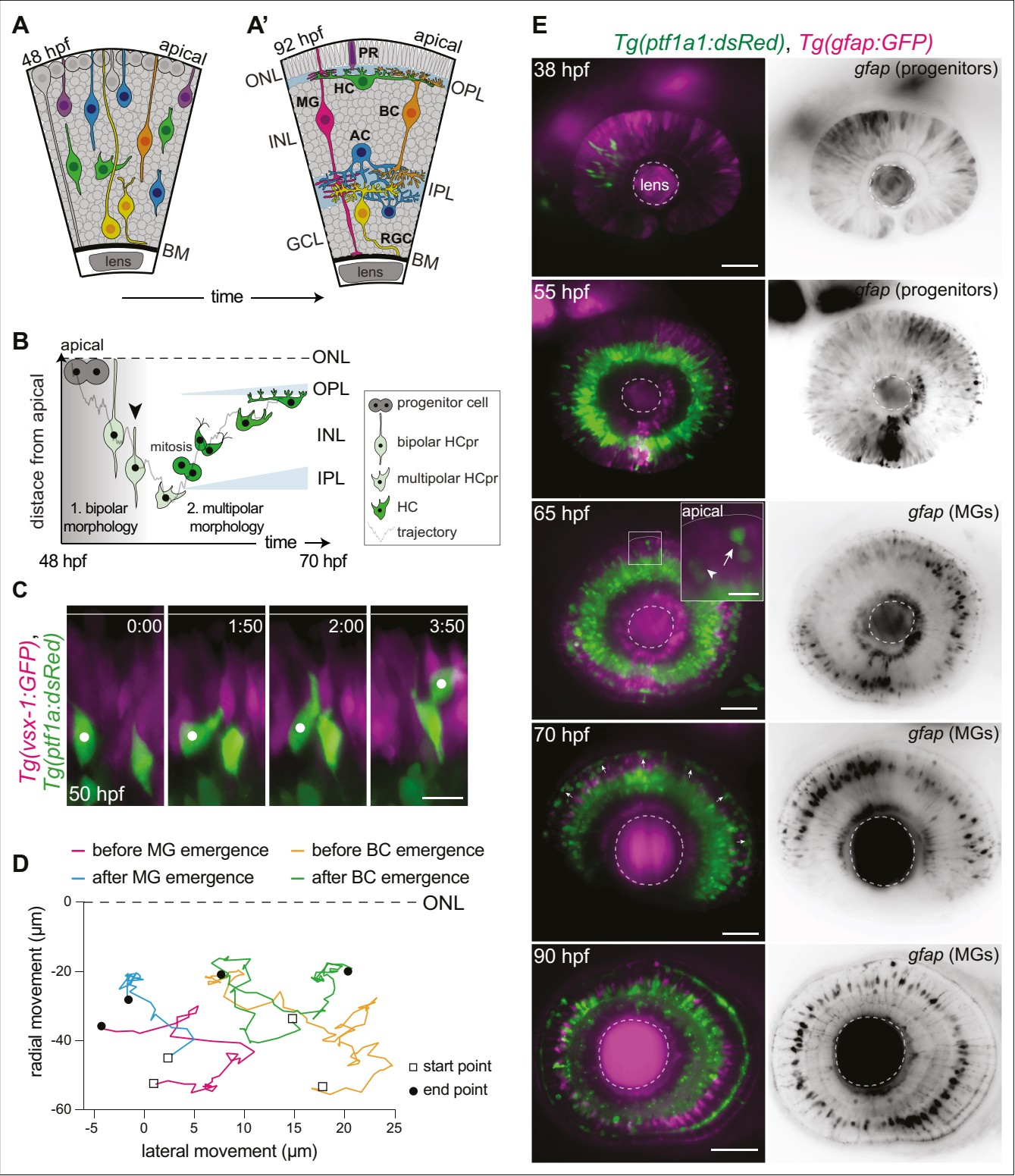

**Figure 1.** Migrating horizontal cells (HCs) do not use radially oriented Müller glia (MGs) or bipolar cells (BCs) for guidance. (**A–A'**) Schematic representation of zebrafish retinal development: (**A**) Neuronal birth and migration (48 hours post fertilization [hpf]). Neuroepithelial progenitors (gray) divide at the apical side and give birth to five major neuron types: photoreceptor (PR), HC, bipolar cell (BC), amacrine cell (AC), retinal ganglion cell (RGC). (**A'**) The layered organization of the mature zebrafish retina (92 hpf). Retinal neurons and a single type of retina glial cell, MGs, are arranged in three nuclear layers from apical to basal: outer nuclear layer (ONL), inner nuclear layer (INL), ganglion cell layer (GCL). These layers are separated by two plexiform layers: outer plexiform layer (OPL) and inner plexiform layer (IPL). The basement membrane (BM) separates the GCL from the vitreous

*Figure 1 continued on next page*

*Figure 1 continued*

body. (**B**) Scheme of bidirectional and bimodal migration of an HC progenitor and its committed precursors. Division of neuroepithelial progenitor (gray) generates HC progenitors (HCprs) (light green). HCprs display a bipolar morphology and are attached to the apical surface during the first phase of migration. Upon apical detachment, HCprs show multipolar morphology. En route to their destination, multipolar HCprs undergo a terminal mitosis to generate two HCs (green), which migrate toward the HC layer. Arrowhead: detachment of HCpr from the apical surface and onset of multipolar migration. Gray line: a representative HC migration trajectory (see also stills and trajectory in *Figure 1—figure supplement 1A-B*). (**C**) Time-lapse of HC tangential migration before BC lamination (50 hpf). *Tg(vsx-1:GFP) labels BCs (magenta) and Tg(ptf1a:DsRed) marks HCs (green)*. White dot: tracked HC. Time in h:min. Scale bar: 10 µm. (**D**) Basal-to-apical migration trajectories of HCs before and after emergence of mature radially oriented BCs and MGs. HCs use both radial (apical-basal) and tangential (lateral) routes to move within the INL. See *Figure 1—source data 1*. (**E**) Maximum projections of retinae at different developmental stages: 38, 55, 65, 70, and 90 hpf. *Tg(ptf1a:dsRed)* labels HCs and ACs (green) and *Tg(gfap:GFP)* labels MGs (magenta). 38 hpf: birth and initiation of HC basal migration. GFAP+ cells are progenitors. 55 hpf: initiation of HC apical migration. 65 hpf: peak of HC apical migration. Higher magnification inset of the outlined region shows an HC at its final destination (arrow), and a migrating HC en route to the apical side (arrowhead). 70 hpf: emergence of the HC layer and MG maturation. arrows: show HCs within the HC layer. 90 hpf: mature and fully laminated retina. Scale bars: 50 µm, inset 65 hpf: 10 µm. See also *Figure 1—figure supplement 1*.

The online version of this article includes the following source data and figure supplement(s) for figure 1:

**Source data 1.** HC migration trajectories.

**Figure supplement 1.** HC migration features are independent of radially oriented cells in the retina.

environment is involved. However, if, how, and to what extent cellular and tissue-wide properties influence HC movements remains unexplored.

To address this question, we here investigated the cellular and tissue-scale parameters that influence HC migration in the developing zebrafish retina. We show that these cells constantly tailor their migration behavior to the limited space within the densely packed retina by frequent and reversible amoeboid-like shape and direction changes. We further uncover that changing organization of the developing retina at the cell or tissue-scale impairs efficient HCpr and HC migration and perturbs proper HC layer formation.

## Results

### HCs do not employ glia-guided migration

HCs lose their apical attachment (*Figure 1—figure supplement 1A-C* , *Video 2*) ~2–4 hr after apical birth and subsequently display multipolar morphology when moving toward their final destination (*Chow et al., 2015*; *Amini et al., 2019*). Since many multipolar interneurons in the neocortex use radially oriented cells featuring bipolar morphology as their migratory scaffold (*Cooper, 2014*), we asked if this also holds true for HCs in the retina. To this end, we examined whether migrating HCs move along BCs or MGs, the two radially oriented retinal cell types with bipolar morphology (*Figure 1A'*).

We performed light-sheet time-lapse imaging using double-transgenic zebrafish embryos *Tg(vsx-1:GFP)* × *Tg(Ptf1-a:dsRed)* labeling BCs and HCs, respectively. The fact that neuronal differentiation in the retina occurs in a wave-like manner (*Hu and Easter, 1999*), allowed us to simultaneously

**Video 1.** Bidirectional trajectory of a tracked horizontal cell (HC) from apical detachment to final positioning. Time-lapse imaging of a typical bidirectional and bimodal migration of an HC. Upon retraction from the apical surface, HC enters a multipolar migration phase. Final mitosis occurs close to the HC layer. *Tg(lhx-1:eGFP)* labels HCs (green), *Tg(Ptf1a:dsRed)* marks amacrine cells (ACs) and HCs (magenta). Red dot: tracked HC; white and blue dots: sister cells of the tracked HC after division. Time in h:min. Scale bar: 5 µm.

https://elifesciences.org/articles/76408/figures#video1

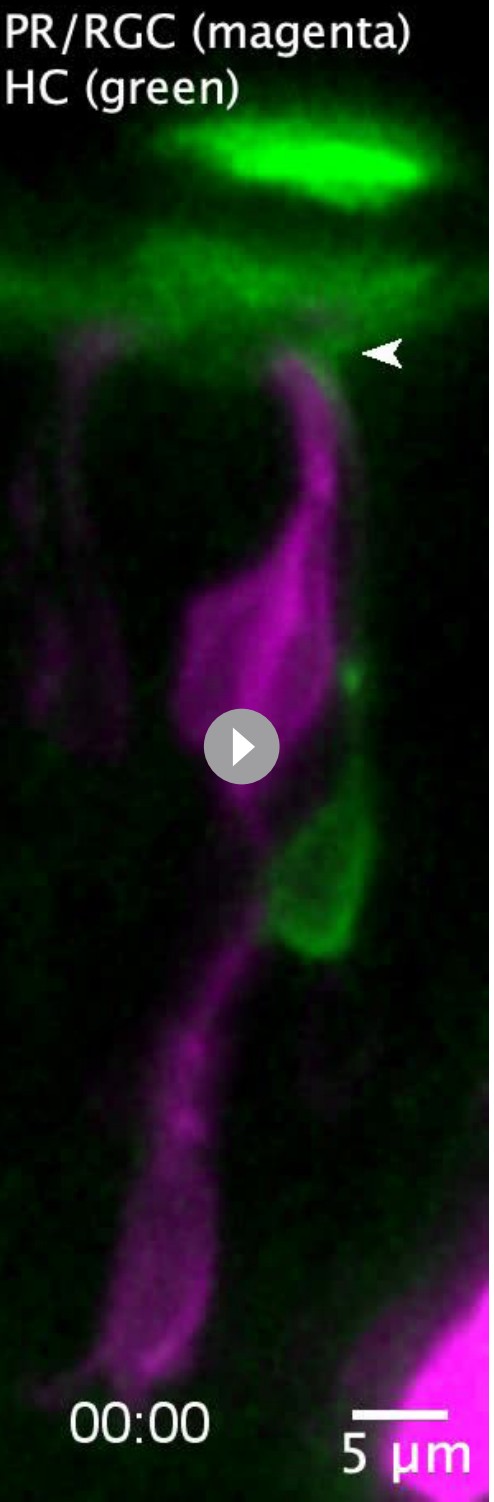

**Video 2.** Retraction of horizontal cell (HC) apical attachment. *Tg(ptf1a:Gal4-VP-16, UAS:gap-YFP)* labels membrane of HCs (green), and *Tg(atoh7:RFP)* labels membranes of photoreceptors (PRs) and retinal ganglion cells (RGCs) (magenta). White arrowhead: apical attachment; red arrowhead: tip of the retracted process. Time in h:min. Scale bar: 5 μm.

https://elifesciences.org/articles/76408/figures#video2

**Video 3.** Tangential migratory track of a horizontal cell (HC) after bipolar cell (BC) lamination. *Tg(vsx-1:GFP)* labels BCs (magenta) and *Tg(ptf1a:DsRed)* marks HCs (green). White dot: tracked HC. Time in h:min. Scale bar: 20 μm.

https://elifesciences.org/articles/76408/figures#video3

visualize retinal regions hosting immature (unlaminated) and mature (laminated) BCs (*Figure 1—figure supplement 1D*). We noted that in regions without laminated BCs, HCs were either already at (*Figure 1—figure supplement 1D*) or en route toward their final position. Further, many HCs did not strictly follow radial migratory trajectories but instead moved in all three dimensions while frequently changing direction (*Figure 1C–D*). The same was seen in regions that hosted laminated BCs (*Figure 1D*, *Video 3*), arguing against steady, direct interaction between radially oriented BCs and migrating HCs.

Similar observations were made when probing a possible association between migrating HCs and

developing MGs, the retinal glial cells. Immunofluorescence stainings of double-transgenic animals *Tg(gfap:GFP)* x *Tg(Ptf1-a:dsRed)* marking MGs and HCs showed that prior to MG emergence at 48 hours post fertilization (hpf), GFAP (glial fibrillary acidic protein) was expressed in radially oriented retinal neurogenic progenitors (*Bernardos and Raymond, 2006*; *Rapaport et al., 2004*; *Figure 1E* – 38 hpf). No GFAP$^+$ cells featuring mature MG morphology at the onset (48 hpf) or peak (55–65 hpf) of HC migration were observed (*Figure 1E* – 55 hpf, *Figure 1—figure supplement 1E* – 50 hpf). When mature MGs emerged (around 65 hpf), some HCs were still en route toward the apical side (*Figure 1E* – 65 hpf, arrowhead), while others had already reached the prospective HC layer (*Figure 1E* – 65 hpf, arrow). From 70 to 72 hpf, when GFAP was specifically expressed in mature MGs (*MacDonald et al., 2015*; *Figure 1E*, *Figure 1—figure supplement 1E*), the majority of HCs were already integrated into the HC lamina (*Figure 1E* – 70 hpf). As seen in embryos expressing BC markers, also in embryos labeled for MGs, HCs followed tangential routes perpendicular to the radial orientation of MG fibers, both before and after MG maturation (*Figure 1D*, *Figure 1—figure supplement 1E'-E''*). Thus, we conclude that the radially oriented BCs or MG fibers are unlikely to be prerequisites for HC movement.

## The INL is densely packed without prominent extracellular matrix components

Cells migrating within tissues can be influenced by mechanical cues from their environment. For example, local gradients in extracellular matrix (ECM) stiffness can guide cell migration in a process termed durotaxis (*Lo et al., 2000*; *Isenberg et al., 2009*; *Roca-Cusachs et al., 2013*; *Bollmann et al., 2015*). In the context of neuronal migration, the ECM can act either as an instructive scaffold along which migration occurs or as a barrier for migrating neurons (*Franco and Müller, 2011*). We asked whether ECM components could influence HC migration, focusing on Laminin α1, a glycoprotein that forms fibrous structures (*Timpl et al., 1979*; *Chung et al., 1979*), and Neurocan (termed here ssNcan in *Tg(ubi: ssNcan-EGFP)*) as a biosensor for hyaluronic acid (HA) (*De Angelis et al., 2017*), a glycosaminoglycan forming hydrogel-like structures (*Grassini et al., 2018*). Our results showed that anti-Laminin α1 and HA were only detected in the basement membrane of the retina at all developmental stages (*Figure 2A-B*, arrowheads) and never within the INL wherein HC migration takes place. A similar localization pattern was observed upon anti-Collagen IV staining (*Figure 2—figure supplement 1A*). Thus, it is unlikely that HCs use ECM scaffolds as a main migratory substrate to reach their final destination.

We next asked whether the developing zebrafish retina features mechanical gradients during HC migration, and if yes, whether and how these gradients change in space and over time. To address this point, we used Brillouin light scattering microscopy, a non-invasive technique which has been recently applied to a broad range of living biological systems including different areas of the CNS (*Girard et al., 2015*; *Schlüßler et al., 2018*, *Scarcelli and Yun, 2012*; *Scarcelli et al., 2011*; *Mattana et al., 2017*). This technique provides information about tissue compressibility by measuring the Brillouin shift values of the sample (*Zhang et al., 2017*; *Prevedel et al., 2019*; *Elsayad et al., 2019b*, *Elsayad et al., 2019a*).

Using a custom-built Brillouin microscopy setup (*Schlüßler et al., 2018*), we profiled the in vivo Brillouin maps of distinct regions of the zebrafish retina (*Figure 2D and E*), at different developmental stages: (1) during HC migration (48 hpf) (*Figure 2C*) and (2) post-HC layer formation (70 hpf) (*Figure 2C'*), in parallel with confocal fluorescence microscopy.

For the Brillouin shift maps we took the longitudinal modulus into consideration as it directly presents the compressibility of the tissue as shown previously (*Schlüßler et al., 2018*). These maps revealed that in comparison to the retinal neuroepithelium, the lens had higher Brillouin shifts at 48 hpf (*Figure 2D'–E'*) which further increased as development progressed (70 hpf) (*Figure 2D''–E''*). At 70 hpf, the retina displays a layered organization composed of five different layers from apical to basal: ONL, OPL, INL, IPL, and GCL (*Figure 2C'*). Notably, Brillouin shift maps of 70 hpf retinae revealed a layered, heterogenous pattern that matched the retinal layers observed in confocal images (*Figure 2D''–E''*). While the two nuclear-free plexiform layers (OPL and IPL) showed lower Brillouin shifts, the three nuclear layers (ONL, INL, GCL) displayed comparably higher Brillouin shifts. This indicated a correlation between Brillouin shift maps and features of each retinal layer. It further revealed that Brillouin shift values could be influenced by nuclear occupation, as shown previously for fibroblasts (*Zhang et al., 2017*). However, Brillouin shift values in the INL showed no obvious differences

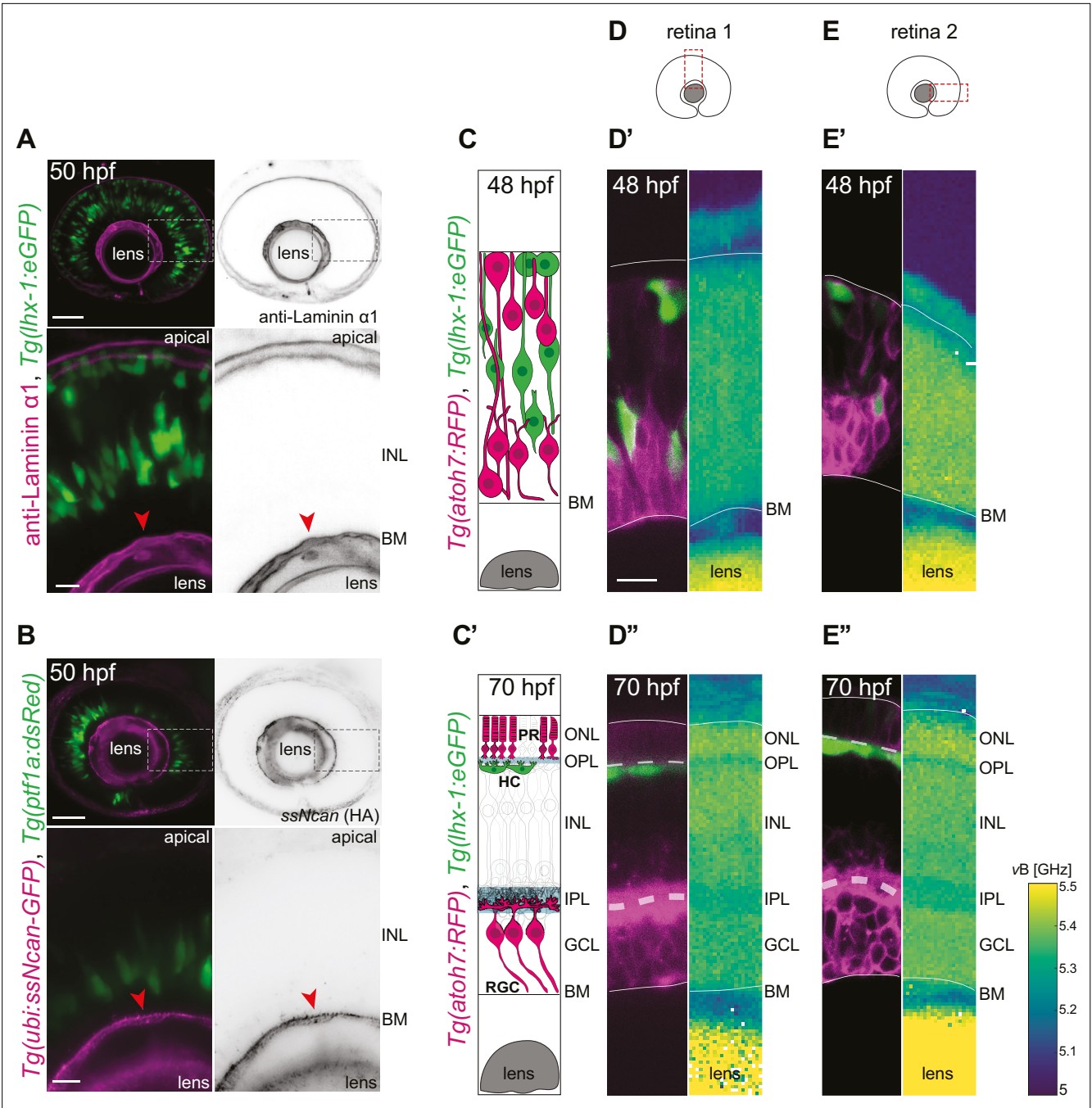

**Figure 2.** The inner nuclear layer (INL) is a densely packed tissue not dominated by extracellular matrix (ECM). (**A–B**) The INL is not dominated by ECM: (**A**) *Tg(lhx-1:eGFP)* labels horizontal cells (HCs) (green). Laminin α1 antibody marks laminin (magenta). (**B**) *Tg(ptf1a:dsRed)* marks HCs and amacrine cells (ACs) (green). *Tg(HA:GFP)* labels hyaluronic acid (HA) (magenta). Higher magnification insets show enrichment of anti-Laminin α1 and HA in the basement membrane (BM) (red arrowheads). Scale bars: 50 μm; insets 10 μm. (**C–C'**) Scheme of structural organization of the retina during development: (**C**) During HC migration at 48 hours post fertilization (hpf); (**C'**) after HC layer formation at 70 hpf. HCs (green); photoreceptors (PRs) and retinal ganglion cells (RGCs) (magenta). (**D–E"**) Brillouin shift maps (right) and their corresponding confocal images (left) of double-transgenic zebrafish of (**D–D"**) retina 1 and (**E-E"**) retina 2. Top: 48 hpf, and bottom: 70 hpf. *Tg(lhx-1:eGFP)* labels HCs and ACs (green), *Tg(atoh7:RFP)* is expressed in RGCs and PRs (magenta). Confocal images were obtained directly after the Brillouin shift measurements. The corresponding Brillouin shift maps of the nuclear layers (outer nuclear layer [ONL], inner nuclear layer [INL], ganglion cell layer [GCL]) show a higher Brillouin shift than the plexiform layers (outer plexiform layer [OPL], inner plexiform layer [IPL]) at 70 hpf. Red dashed boxes in D–E: imaged regions. Scale bar: 10 μm. See also *Figure 2—figure supplement 1*.

The online version of this article includes the following source data and figure supplement(s) for figure 2:

**Source data 1.** *Figure 2—figure supplement 1C''*.

**Figure supplement 1.** No collagen is found in the densely packed INL.

along the apico-basal axis, during (*Figure 2D'–E'*) and after HC migration (*Figure 2D"–E"*). Thus, HC migration seems to occur in a homogenous environment without notable local compressibility gradients.

The fact that our Brillouin shift values were higher in regions with high cell body density compared to the nuclei-free plexiform layers as also previously reported (*Sánchez-Iranzo et al., 2020*) made us ask about cell packing in the INL at stages when HC migration is at its peak (50–65 hpf). To this end, we performed live imaging of double-transgenic zebrafish embryos *Tg(lhx-1:eGFP)* × *Tg(βact-in:mKate2-ras)* labeling HCs and membrane of all retinal cells, respectively. No extracellular space between neighboring cells within the INL was detected during periods of HC migration (*Figure 2—figure supplement 1B-B'*). Confocal imaging of cell membrane (*Tg(βactin:mKate2-ras)*) and nuclear envelope (*Tg(βactin:eGFP-lap2b)*) further showed no detectable space between neighboring INL cells as nuclei took up the majority of the cell space (*Figure 2—figure supplement 1C-C'"*). Thus, similar to the neuroepithelial stage (*Matejčić et al., 2018*), also during neurogenesis, the developing INL seemed to contain closely packed cells but no detectable ECM components. This suggested that HCs migrate through a physically constraining cell-dense tissue with no preexisting path.

## Migrating HCs show cell and nuclear deformations

We showed that migrating HCs navigate within a crowded environment with no or little ECM scaffolding structures (*Figures 1C, E and 2A–B*, *Figure 2—figure supplement 1A*)**,** or compressibility gradients (*Figure 2D–E"*). Two major strategies have been shown to be employed by cells migrating in crowded environments with physical constraints: (1) active generation of migratory tracks by protease-dependent local ECM degradation or ECM remodeling (e.g. used by mesenchymal cells such as fibroblasts during wound healing, and aggressively migrating tumor cells) (*Wolf et al., 2007*; *Wolf and Friedl, 2011*; *Krause and Wolf, 2015*), or (2) amoeboid adaptations of cell shape, migration path, and/or direction to the limited available space (e.g. seen for leukocytes like neutrophils and dendritic cells) (*Lammermann and Sixt, 2009*; *Friedl and Wolf, 2010*; *Yamada and Sixt, 2019*). The INL is a cell-dense packed tissues environment wherein no major ECM components were detected (*Figure 2A–B*, *Figure 2—figure supplement 1A*). To test whether ruptures in the tissue existed, which could hint toward protease activities of the migrating cells, tissue integrity was probed using double-transgenic zebrafish retinae *Tg(lhx-1:eGFP)* × *Tg(βactin:mKate2-ras)* labeling HCs and PRs, and the membranes of all retinal cells, respectively. We did not detect any tissue ruptures or holes, neither at the peak of HC migration nor after HC migration (*Figure 2—figure supplement 1B-B'*). Thus, protease-dependent path generation to move through the crowded retina might not a major component in this context.

Typically, amoeboid migrating cells undergo cytoplasmic and nuclear deformations which allow them to navigate through crowded environments in vivo (*Friedl et al., 2011*, *Wolf et al., 2003*; *Salvermoser et al., 2018*; *Manley et al., 2020*) or in narrow channels in vitro. We thus asked whether such deformations accompanied and/or influenced HC migration behavior. To achieve mosaic cell labeling, we either used blastomere transplantation of *Tg(lhx-1:eGFP)* to mark cell bodies or co-injected trbeta2:tdTomato and LAP2b:eGFP DNA constructs to visualize cell bodies and nuclear envelopes of HCs, respectively. We observed that migrating HCs exhibited a wide variety of cellular and nuclear shape changes, ranging from elongated to bended (*Figure 3A–A"*, *Figure 3—figure supplement 1A-A"*, *Figure 3—figure supplement 1E-E'*).

To quantify these morphological alterations, we used the open-source Icy platform (*de Chaumont et al., 2012*) (http://icy.bioimageanalysis.org) for automated segmentation of HC cell and nucleus contours (*Manich et al., 2020*; *Figure 3A'–A"*, *Figure 3—figure supplement 1A-A"*, *Video 4*). We measured perimeter (μm), elongation ratio and sphericity (%) of cell body and nucleus of HCs during migration, as well as during their final mitosis (*Figure 3B*), on their basal-to-apical journey. At mitosis, HCs displayed an elongation ratio of ~1 (*Figure 3D*) and a sphericity of ~100% (*Figure 3E*). Thus, cell bodies and nuclei of mitotic HCs adopt a spherical shape (*Figure 3A'–A"*, *Figure 3—figure supplement 1A-A"*), as seen for most animal cells entering mitosis (*Taubenberger et al., 2020*). During migration, HCs exhibited a more elongated cellular morphology and underwent multiple cellular deformations (*Figure 3A'*, *Figure 3—figure supplement 1A-A"*) as was confirmed by frequent changes in their perimeter (μm), elongation ratio, and sphericity (%) (*Figure 3C–E*, *Figure 3—figure supplement 1B-D*, *Video 4*). These cell-shape changes were accompanied by alterations in nuclear

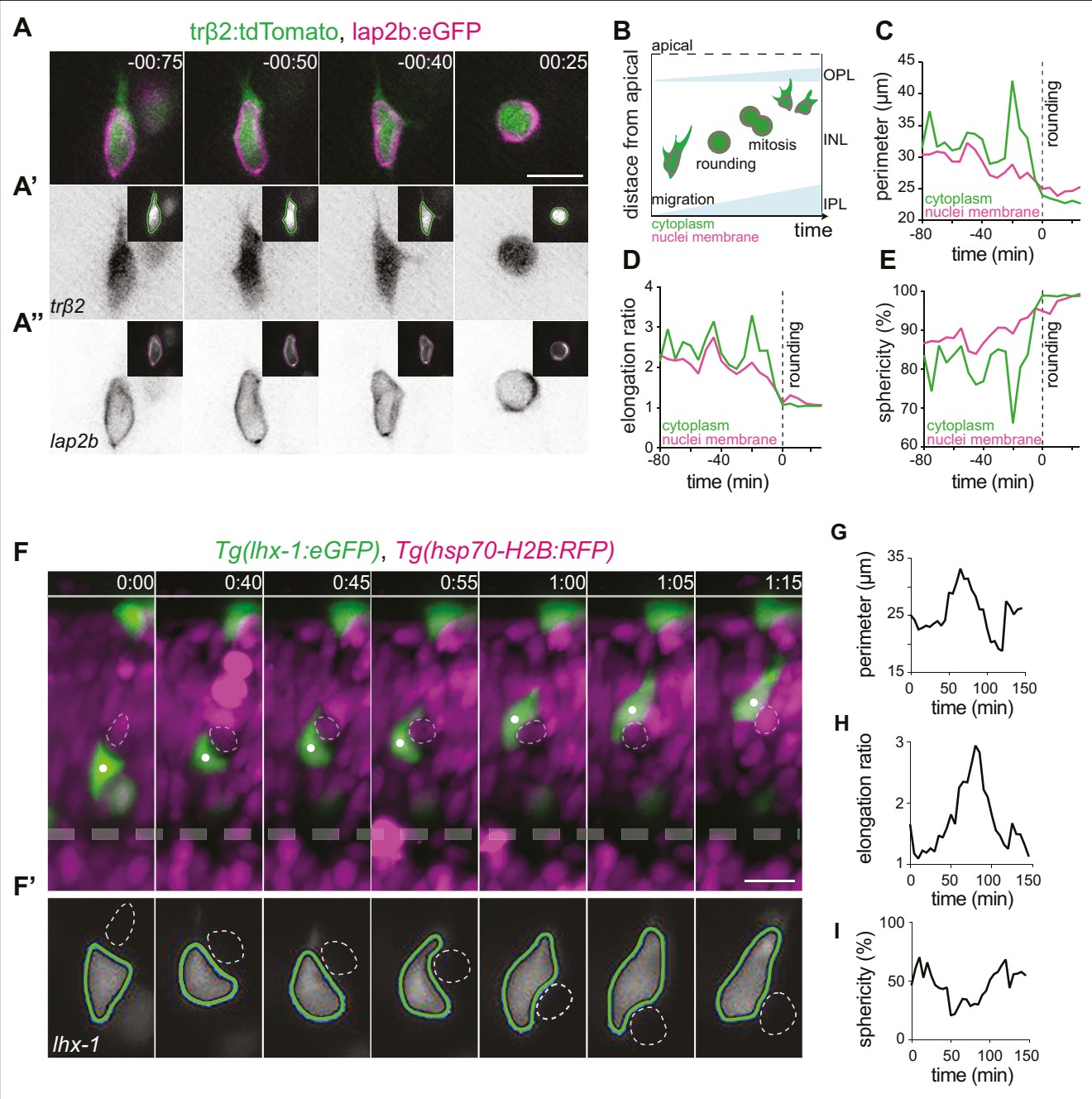

**Figure 3.** Horizontal cells undergo cell and nuclear deformations as they migrate through the crowded inner nuclear layer (INL). (**A–E**) The cell bodies and nuclei of migrating horizontal cells (HCs) are deformable. (**A**) Time-lapse of an HC during migration and at entry into mitosis. The HC cell body is visualized by trβ2:tdTomato (green), and its nuclear envelope by lap2b:eGFP (magenta) DNA constructs. Insets show the corresponding segmented contours of the tracked HC's generated by Icy: (**A'**) cell body (green lines) and (**A''**) nucleus (magenta lines) which were used to extract cellular and nuclear morphodynamic features in (**C–E**). Scale bar: 10 μm. (**B**) Schematic representation of a typical basal-to-apical migration trajectory of an HC progenitor which undergoes mitosis en route. During mitosis, HCs switch from an elongated shape into a spherical morphology. (**C–E**) Graphs show quantification of the dynamics of the cell and nucleus shape changes of the migrating HC depicted in (**A**): (**C**) perimeter (μm), (**D**) elongation ratio, and (**E**) sphericity (%). Note that the minimal perimeter (μm), minimal elongation ratio, and maximal sphericity (%) are reached upon cell rounding during mitosis. Dashed lines: onset of HC rounding. See *Figure 3—source data 1*. (**F–I**) Migrating HCs squeeze in the crowded retina to overcome local physical obstacles. (**F**) Stills from light-sheet time-lapse imaging show that the migrating HC (green) undergoes cell-shape deformations to circumvent a mitotic neighboring cell (dashed circle). *Tg(lhx-1:eGFP)* labels HCs (green), *Tg(hsp70-H2B:RFP)* marks nuclei of all retinal cells (magenta). White dot: tracked HC; line: apical surface; dashed line: inner plexiform layer (IPL). Scale bar: 20 μm. (**F'**) The automated segmented contours of the cell body of the migrating HC (green line) generated by Icy. Dashed circle: the nucleus of the neighboring cell from (**F**). (**G–I**) Graphs represent quantifications of cell

*Figure 3 continued on next page*

*Figure 3 continued*

morphodynamic changes in HC from (**F–F'**): (**G**) perimeter (μm), (**H**) elongation ratio, and (**I**) sphericity (%). Time in h:min (**A, F**). See *Figure 3—source data 1*. See also *Figure 3—figure supplements 1 and 2*.

The online version of this article includes the following source data and figure supplement(s) for figure 3:

**Source data 1.** *Figure 3* Panel D-E; and G-I.

**Source data 2.** *Figure 3—figure supplement 1B-D*.

**Source data 3.** *Figure 3—figure supplement 2B and D*.

**Figure supplement 1.** Analysis of cell and nuclear deformations of HC during migration.

**Figure supplement 2.** Analysis of HC migration behaviour upon onstacle encounter.

perimeter (μm), elongation ratio, and sphericity (%) (*Figure 3A" and C–E*) and dynamic indentations of the nucleus (*Figure 3—figure supplement 1E-E'*, *Video 5*).

To test whether a correlation between encountered tissue obstacles and HC shape and direction changes existed, we monitored HC migration (*Tg(lhx-1:eGFP)*) in relation to the surrounding local environment (*Tg(hsp70-H2B:RFP)*). We noted that in some cases migrating HCs featured cellular deformations when encountering a neighboring cell that entered mitosis (*Figure 3F–I*, *Video 6*). However, HC direction and shape changes do not only occur upon encountering mitotic cells but also when the tissue seemed impassable. In this case, HCs changed both their shape and direction of migration (*Figure 3—figure supplement 2A-A',C-C'*), often taking less direct routes to their final position (*Figure 3—figure supplement 2B,D*, ). Notably, the adaptive cellular and nuclear deformations shown by HCs seemed reversible after circumventing the encountered physical barrier (*Figure 3F–I*, *Figure 3—figure supplement 2*).

Together, these data make us speculate that migrating HCs tailor their cellular and nuclear shapes, paths and directions to their cell-dense

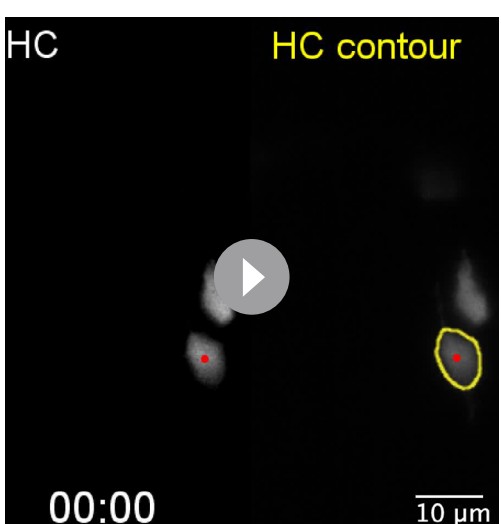

**Video 4.** Horizontal cells (HCs) undergo frequent and reversible cell morphodynamic changes during migration. Left: Time-lapse imaging shows basal-to-apical migration of an HC which was transplanted from *Tg(lhx-1:eGFP)* transgene embryo into wild-type (WT) embryos. HC rounds at the apical side (last frame). Right: Representative sequences of the automated segmented contours of the cell body (yellow line) generated by Icy. Red dot: tracked HC. Time in h:min. Scale bar: 10 μm.

https://elifesciences.org/articles/76408/figures#video4

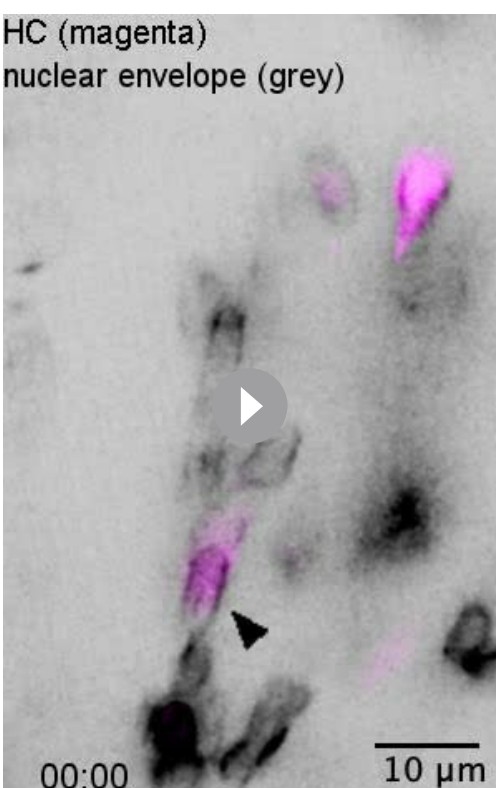

**Video 5.** Nuclear deformations during horizontal cell (HC) migration. Time-lapse imaging of an HC shows that it undergoes nuclear deformations while squeezing through the crowded retina. *βactin:lap2b-eGFP* labels nuclear envelopes (gray), *trβ2:tdTomato* is expressed in HCs (magenta). Black arrowhead: tracked HC. Time in h:min. Scale bars: 10 μm.

https://elifesciences.org/articles/76408/figures#video5

HC/AC

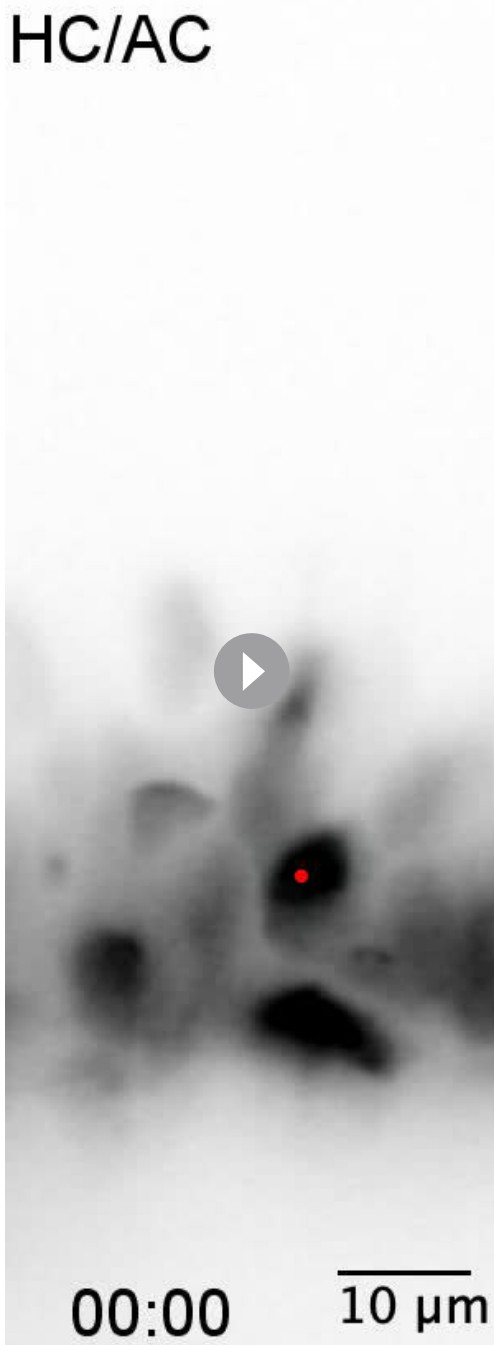

**Video 6.** Horizontal cell (HC) undergoes cell-shape deformations to bypass its neighboring cell. Time-lapse imaging of an HC migration from basal-to-apical inner nuclear layer (INL). *Tg(Ptf1a:dsRed)* labels HCs and amacrine cells (ACs). Red dot: tracked HC. Time in h:min. Scale bar: 10 µm.

https://elifesciences.org/articles/76408/figures#video6

HC membrane

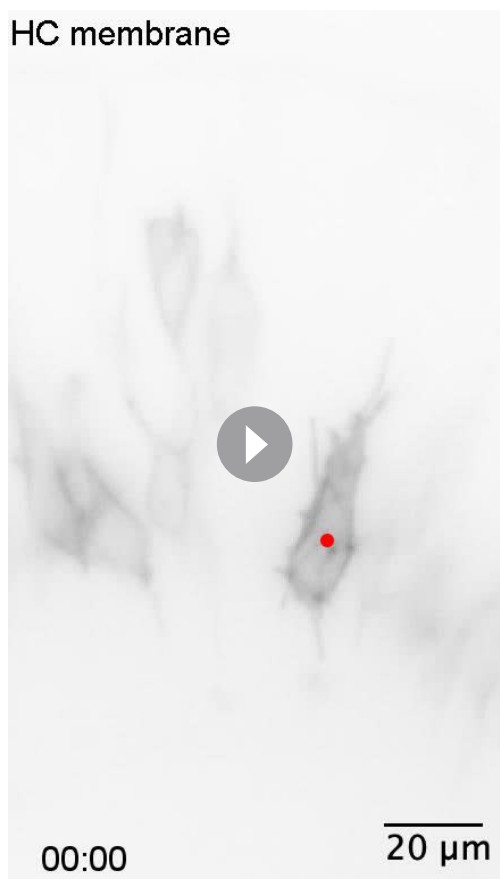

**Video 7.** Horizontal cells (HCs) send dynamic protrusions during migration and only form blebs prior to mitosis. Time-lapse imaging of an HC migration shows that it sends dynamic protrusions while migrating. Upon rounding prior to mitosis, the tracked HC displays membrane blebbing. *Tg(ptf1a:Gal4-VP-16,UAS:gap-YFP)* labels HC membrane. Red dot: tracked HC, red arrowhead: blebs, white and black dots: sister cells of the tracked HC after division. Time in h:min. Scale bar: 20 µm.

https://elifesciences.org/articles/76408/figures#video7

surrounding tissue environment. This behavior could serve as a space adaptation strategy enabling HCs to successfully move within the densely packed and dynamically complex retina. Many types of amoeboid migration from *Dictyostelium* to leukocytes (*Friedl and Wolf, 2010*; *Trepat et al., 2012*; *Arts et al., 2021*) show similar morphological changes. Therefore, our results suggest that HCs can undergo amoeboid-like migration.

## HCs display a polarized front-rear morphology

To examine whether HC migration displayed further cell biological characteristics of amoeboid migration, we investigated HC protrusion activity. Classically, two types of protrusions guide amoeboid migration: (1) amoeboid blebs (e.g. seen in macrophages) (*Yoshida and Soldati, 2006*; *Charras and*

*Paluch, 2008*; *Bergert et al., 2012*) and (2) amoeboid pseudopods (e.g. used by *Dictyostelium* and neutrophils) (*Trepat et al., 2012*; *Friedl and Wolf, 2010*). To elucidate HC protrusion activity, we monitored HC membranes in *Tg(ptf1a:Gal4-VP-16,UAS:gap-YFP)* embryos. As seen for many other cell types (*Boss, 1955*; *Burton and Taylor, 1997*; *Fishkind et al., 1991*; *Boucrot and Kirchhausen, 2007*), HCs displayed membrane blebs during mitosis but never during migration (*Figure 4—figure supplement 1A*, *Video 7*).

In contrast, we found that HCs feature multiple dynamic protrusions with different directionality as they moved sideways (tangentially), up (toward the apical), or down (toward the basal) within the INL. These protrusions showed different thicknesses, lengths, morphologies (branched vs. unbranched), and orientations, and exhibited dynamic extension and retraction from the cell soma in multiple directions (*Figure 4A*, *Video 7*). At times, two or multiple protrusions were seen to extend from the cell body at the cell front (*Figure 4A*, *Figure 4—figure supplement 1B*), another feature also reported for amoeboid migrating cells (*Weber et al., 2013*; *Renkawitz et al., 2019*; *Kameritsch and Renkawitz, 2020*).

To analyze the overall orientation of protrusions with respect to the direction of HC movement, tips of each protrusion were manually tracked in individual HCs (n=6) (*Figure 4A–C'*, *Figure 4—figure supplement 1F-I'''*, *Video 8*). The angle between each protrusion and direction of instantaneous cell movement was then quantified (*Figure 4B*) throughout the tracked basal-apical migration of each HC. This analysis revealed that the overall probability of finding protrusions with orientations parallel to the direction of cell movement was higher than finding protrusions pointing in other directions (*Figure 4C*, *Figure 4—figure supplement 1F'-I'*). This implied that HCs show some front-back polarity while migrating. Moreover, we observed that the protrusions were typically more randomly oriented when HCs moved with lower speed (cell speeds below 25 percentile of the measured speeds of that cell during its tracked trajectory) (*Figure 4C'*, *Figure 4—figure supplement 1F"-I"*) than when cells moved with high speed (cell speeds above 75 percentile of the measured speeds of that cell during its tracked trajectory). Overall, the frequency of protrusions with 0° (cell-front) and 180° (cell-back) was higher at higher speeds (*Figure 4C"*, *Figure 4—figure supplement 1F"'-I'''*).

One possible component that could be responsible for front-rear polarity in HCs during migration is the asymmetric distribution of stable F-actin as reported for amoeboid migrating cells including leukocytes (*Cassimeris et al., 1990*), dendritic cells in 3D (*Insall and Machesky, 2009*; *Lammermann et al., 2008*), neutrophils (*Yoo et al., 2010*; *Manley et al., 2020*; *Barros-Becker et al., 2017*), and neutrophil-like cells in vitro (*Cooper et al., 2008*). We thus investigated F-actin distribution in HCs using two distinct actin bioprobes: Lifeact (17 amino acids of yeast Abp140), which labels all filamentous F-actin structures (*Yoo et al., 2010*; *Fritz-Laylin et al., 2017*), and utrophin (calponin homology domain) (Utr-CH) that has been shown to preferentially bind to a more stable cortical population of F-actin (*Burkel et al., 2007*; *Riedl et al., 2008*; *Yoo et al., 2010*; *Belin et al., 2014*; *Barros-Becker et al., 2017*). We found that while Lifeact was observed in the cell soma and protrusions (*Figure 4D*, *Video 9*), Utr-CH was absent from the cell soma, membrane-proximal regions, and protrusions (*Figure 4E*, *Video 10*). Instead, measurements of Utr-CH fluorescent intensity profiles along the cell axis showed its enrichment at the back of the migrating HC (*Figure 4D and F'*, *Figure 4—figure supplement 1E*). This is akin to the structure seen at the cell rear of amoeboid migrating cells also known as uropod (*Barros-Becker et al., 2017*; *Manley et al., 2020*; *Hind et al., 2016*).

Monitoring Utr-CH during HC migration showed that the uropod-like structure underwent dynamic changes of extension and retraction (*Figure 4—figure supplement 1C-D*, *Video 10*), a feature previously reported for amoeboid migrating neutrophils (*Manley et al., 2020*).

Front-back polarity manifestation of migrating cells is typically accompanied by asymmetric positioning of organelles, including centrosomes (*Kupfer et al., 1982*; *Luxton and Gundersen, 2011*). We thus monitored the distribution and dynamics of centrosomes during HC migration, using Centrin:GFP (centrosome marker) and *trβ2:tdTomato* (cytoplasmic HC marker) DNA injections (*Figure 4G*, *Figure 4—figure supplement 2A-A'*). Our analysis revealed that HC centrosomes displayed a highly variable and dynamic localization and continuously shifted their positions from the cell-front to the cell-back while occasionally staying in the cell's middle (*Figure 4G–H*, *Figure 4—figure supplement 2B-D*). This oscillating configuration suggested that centrosome position did not directly influence the direction of HC movement.

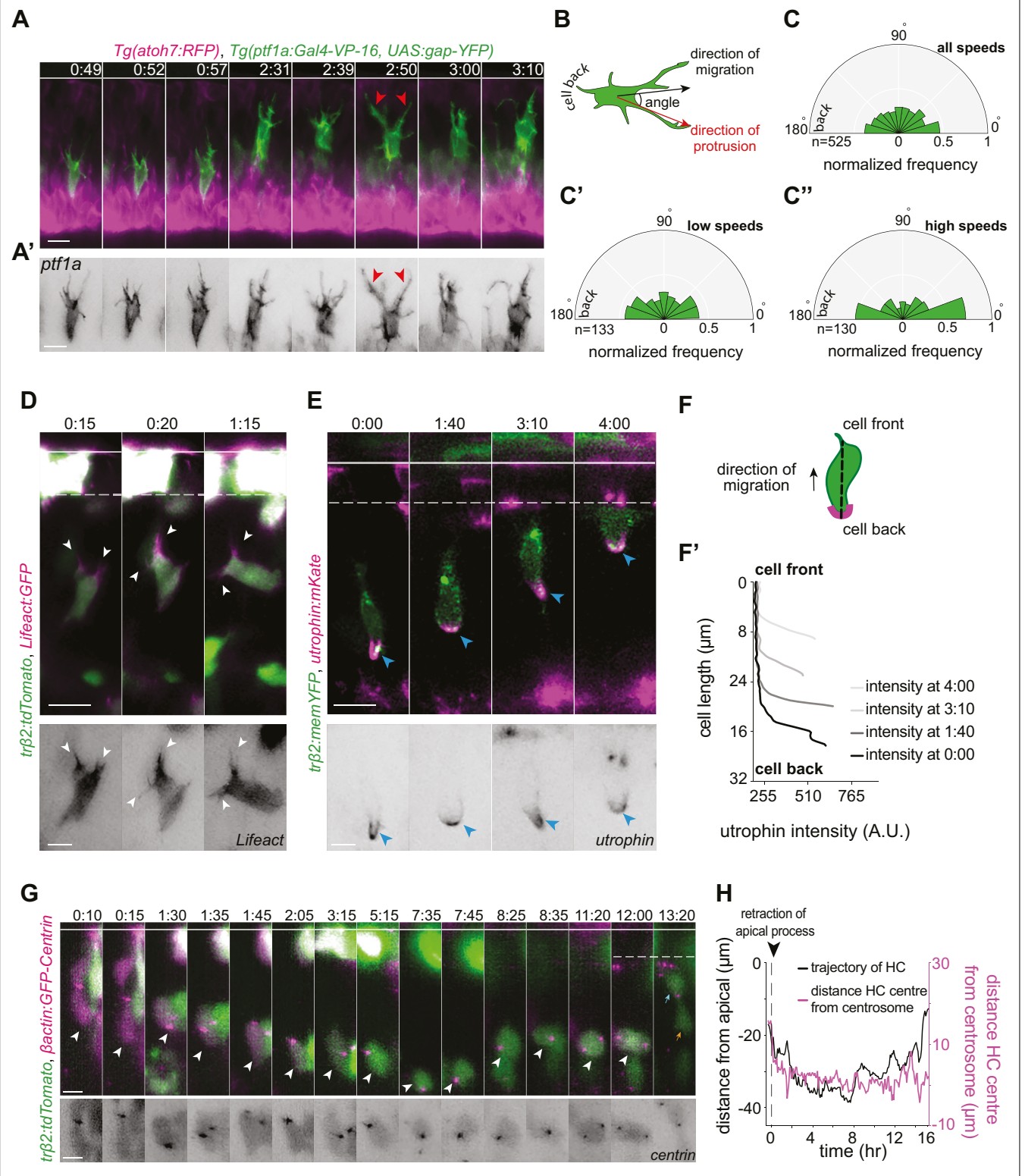

**Figure 4.** Horizontal cells feature hallmarks of protrusion-based amoeboid-like migration. (**A**) Time-lapse sequence of highly dynamic membrane protrusions pointing in different directions (red arrowheads) in a migrating horizontal cell (HC). *Tg(ptf1a:Gal4-VP-16,UAS:gap-YFP)* labels membrane of HCs (green), *Tg(atoh7:RFP)* marks retinal ganglion cells (RGCs) and photoreceptors (PRs). Time interval = 1 min. Scale bar: 50 μm. (**A'**) Higher magnification insets of HC cell membrane from (**A**). Scale bar: 5 μm. (**B**) Scheme of protrusion angles measurements. (**C–C''**) The frequency distribution of the angle between the direction of instantaneous movement of HC and its protrusions; (**C**) throughout the tracked basal-to-apical journey (t=4 hr),

*Figure 4 continued on next page*

*Figure 4 continued*

(**C'**) at low speeds: speeds below 25 percentile of this HC's speed (<0.38 μm/min), (**C''**) at high speeds: above 75 percentile of this HC's speed (>0.93 μm/min). Minimum and maximum speeds observed for this cell are 0.06 and 4.44 μm/min, respectively. A protrusion pointing exactly toward the direction of cell movement exhibits an angle of 0°, and a protrusion pointing exactly opposite exhibits an angle of 180°. The radius indicates the normalized frequency for each angle bin, that is, the number of frames observed with the angle belonging to the particular angle bin for the given velocity condition normalized by the total number of frames observed in the given velocity condition. All measurements are from the migrating HC in (**A–A'**). See *Figure 4—source data 1*. (**D–F'**) F-actin distribution in migrating HCs: (**D**) trβ2:tdTomato and Lifeact-GFP DNA constructs mark HCs (green) and all filamentous F-actin (magenta), respectively. Lifeact-GFP is detected in the leading protrusions (white arrowheads) and the cell cortex. Scale bar: 20 μm. Bottom: close-up of Lifeact-GFP (gray) from (**D**). Scale bar: 5 μm. (**E**) trβ2:memYFP is expressed in HCs (green) and utrophin-mKate marks the stable filamentous F-actin (magenta). Utrophin is enriched at the cell back (blue arrowheads). Scale bar: 20 μm. Bottom: high magnification of utrophin:mKate (gray) in migrating HC from (**E**). Scale bar: 5 μm. (**F**) Scheme of line scan measurements of utrophin:mKate fluorescence intensity. (**F'**) Average utrophin:mKate fluorescent intensity profile of images at (**E**). Cell front and back are determined by the direction of HC movement. See *Figure 4—source data 2*. (**G–H**) Time series showing the dynamics of centrosome position in HCs from birth to final positioning. (**G**) β-actin:GFP-Centrin and trβ2:tdTomato DNA plasmids label centrosomes (magenta) and HCs (green), respectively. White arrowhead: HC. Cyan and orange arrows: sister HCs after mitosis. Bottom panel shows close-up of β-actin:GFP-Centrin (gray) in the tracked HC. White line: apical surface; dashed white line: HC layer. Scale bar: 20 μm. (**H**) Graph showing migration trajectory of the represented HC (black line) and the distance between its centrosome and center (magenta line) throughout migration. Arrowhead: time of detachment of the apical process. See *Figure 3—source data 1*, *Figure 3—source data 2*, *Figure 3—source data 3*. Time in h:min (**A–G**). See also *Figure 4—figure supplements 1 and 2*.

The online version of this article includes the following source data and figure supplement(s) for figure 4:

**Source data 1.** *Figure 4c-c''*; *Figure 4—figure supplement 1F-I'*.

**Source data 2.** *Figure 4F*; *Figure 4—figure supplement 1D-E*.

**Source data 3.** Source data for *Figure 4H* and *Figure 4—figure supplement 2B-D*.

**Figure supplement 1.** Analysis of HC protrusions.

**Figure supplement 2.** Analysis of HC centrosome position.

Overall, we concluded that similarly to amoeboid moving cells, HCs acquire a polarized morphology with persistent rearward polarization of stable F-actin, most likely without contribution of centrosome position. This further supports the idea that HCs employ amoeboid-like migration strategies to migrate in the zebrafish retina.

## Tissue-wide overexpression of Lamin A impacts the efficiency of HC migration and lamination

We showed that HCs undergo frequent cellular and nuclear deformations (*Figure 3A–D*, *Figure 3—figure supplements 1–2*) while migrating through the densely packed INL (*Figure 2—figure supplement 1B-C''*). Consequently, we asked whether changing the properties of nuclei as the biggest and bulkiest cell organelle (*Martins et al., 2012*; *Lammerding, 2011*) could impact HC migration. Some nuclear properties are determined by the differential expression of type V intermediate filament proteins of A- and B-type lamins which are

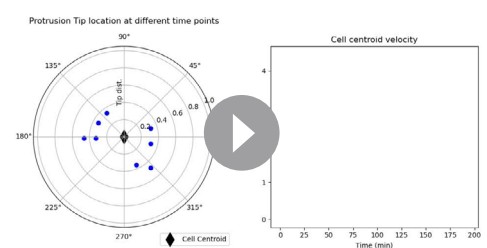

**Video 8.** Angle distribution. The frequency distribution of the angle between the direction of instantaneous horizontal cell (HC) movement and its protrusions. The radius indicates the normalized frequency for each angle bin.
https://elifesciences.org/articles/76408/figures#video8

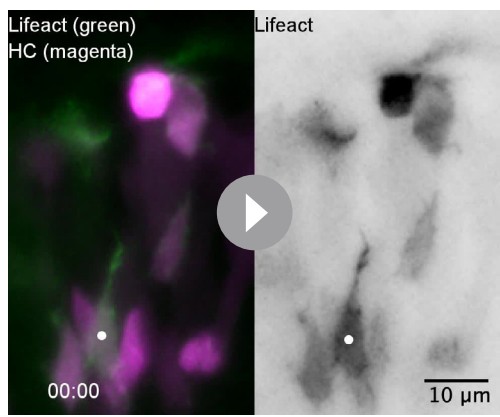

**Video 9.** Lifeact-GFP is detected at the leading protrusions and the cell cortex of migrating horizontal cells (HCs). Left: trβ2:tdTomato marks HCs (green) and Lifeact-GFP labels all filamentous F-actin (magenta). Time in h:min. Scale bar: 10 μm.
https://elifesciences.org/articles/76408/figures#video9

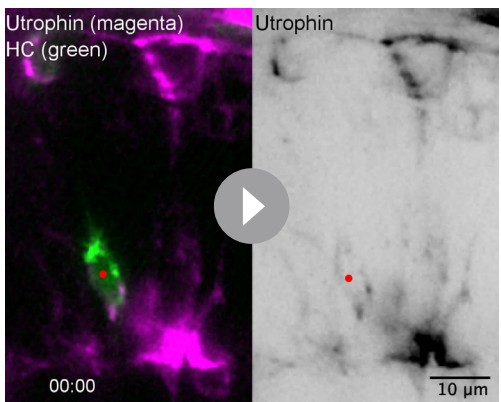

**Video 10.** Migrating horizontal cells (HCs) display a front-rear utrophin distribution. trβ2:memYFP is expressed in HCs (green) and utrophin-mKate marks the stable filamentous F-actin (magenta).Time in h:min. Scale bar: 10 μm.
https://elifesciences.org/articles/76408/figures#video10

part of the nuclear lamina (*Broers et al., 1997*; *Gruenbaum et al., 2005*; *Lammerding et al., 2006*; *Gerace and Huber, 2012*; *Burke and Stewart, 2013*). Particularly A-type lamins (A, C, and C2) were shown to be inversely correlated to the deformability of the nucleus as increasing their expression levels has been linked to decreasing nuclear deformability (*Lammerding et al., 2006*; *Harada et al., 2014*; *Rowat et al., 2013*; *Swift et al., 2013*; *McGregor et al., 2016*).

Nuclei in the developing zebrafish retina only express negligible levels of A-type lamins (*Yanakieva et al., 2019*). Thus, we probed whether and how increasing the expression of Lamin A (LMNA) at the tissue scale could influence HC migration. To test this, we generated a zebrafish transgenic line *Tg(hsp70:LMNA-mKate2)* in which LMNA overexpression in all cells is induced upon heat-shock. We used heat-shock conditions upon which retinal development was not stalled or impaired neither in control nor in *Tg(hsp70:LMNA-mKate2)* animals (Materials and methods). We then quantitatively analyzed HC migration and layer formation in fixed samples of *Tg(lhx-1:eGFP)* (as controls) (*Figure 5A*), and *Tg(lhx-1:eGFP)* x *Tg(hsp70:LMNA-mKate2) double-transgene retina*e (*Figure 5B*), 40 hr after heat-shock at 90 hpf. At this stage, HC layer formation is complete in physiologically unperturbed embryos. Similarly, in control heat-shocked retinae, all HCs were positioned within the HC layer at 90 hpf (*Figure 5A*), showing that heat-shock treatment of the embryos does not impair HC migration and layer formation. In contrast, in the LMNA overexpressing retinae, many HCs were found at ectopic basal positions, mostly within the INL and at times even in the GCL at 90 hpf (*Figure 5B*, arrowheads).

To assess whether the apical migration of the ectopically positioned HCs is delayed or abrogated as a result of tissue-level LMNA overexpression, we performed long-term in vivo imaging (40–45 hr) of *Tg(lhx-1:eGFP)* × *Tg(hsp70:LMNA-mKate2) double-transgene embryos* and monitored the overall HC layer formation until 92 hpf. We found that while the majority of HCs in LMNA overexpressed conditions reached the HC layer at 92 hpf, a subset of HCs remained at ectopic basal positions (*Figure 5C*). Analyzing the migration trajectories of tracked HCs revealed that in tissue-scale LMNA overexpressed conditions, many HCs failed to move back toward the apical INL (*Figure 5D–E*, *Figure 5—figure supplement 1B*) and consequently resided in ectopic basal positions away from their final position (*Figure 5—figure supplement 1C*).

In *Tg(hsp70:LMNA-mKate2)* animals, LMNA is overexpressed in both HCs and their surrounding cells upon heat-shock. To dissect a relative contribution of elevating LMNA levels of the surrounding tissue environment on HC migration efficiency, we injected *hsp70:LMNA-mKate2* DNA constructs to mosaically overexpress LMNA. LMNA overexpressed HCs surrounded by cells not expressing LMNA migrated very similarly to controls and successfully reached the HC layer (*Figure 5—figure supplement 1A*). Thus, while mosaic LMNA overexpression of HCs alone does not impede HC migration and layer formation, changing the nuclear laminar composition at the tissue scale hampers the migration efficiency of HCs causing less efficient HC migration and leading to cells never reaching their final position.

## The IPL acts as a barrier for HC migration

We wondered whether HC migration and efficiency could be impacted by physical barriers in the tissue. In the developing zebrafish retina, five layers with discrete properties (*Figure 1A*) and topographical features emerge during neuronal lamination. We previously showed that the IPL, once formed, negatively influences the depth of HC migration, as HCs did not basally pass it (*Amini et al., 2019*). This suggested that the IPL, despite being devoid of cell bodies, may act as a non-passable

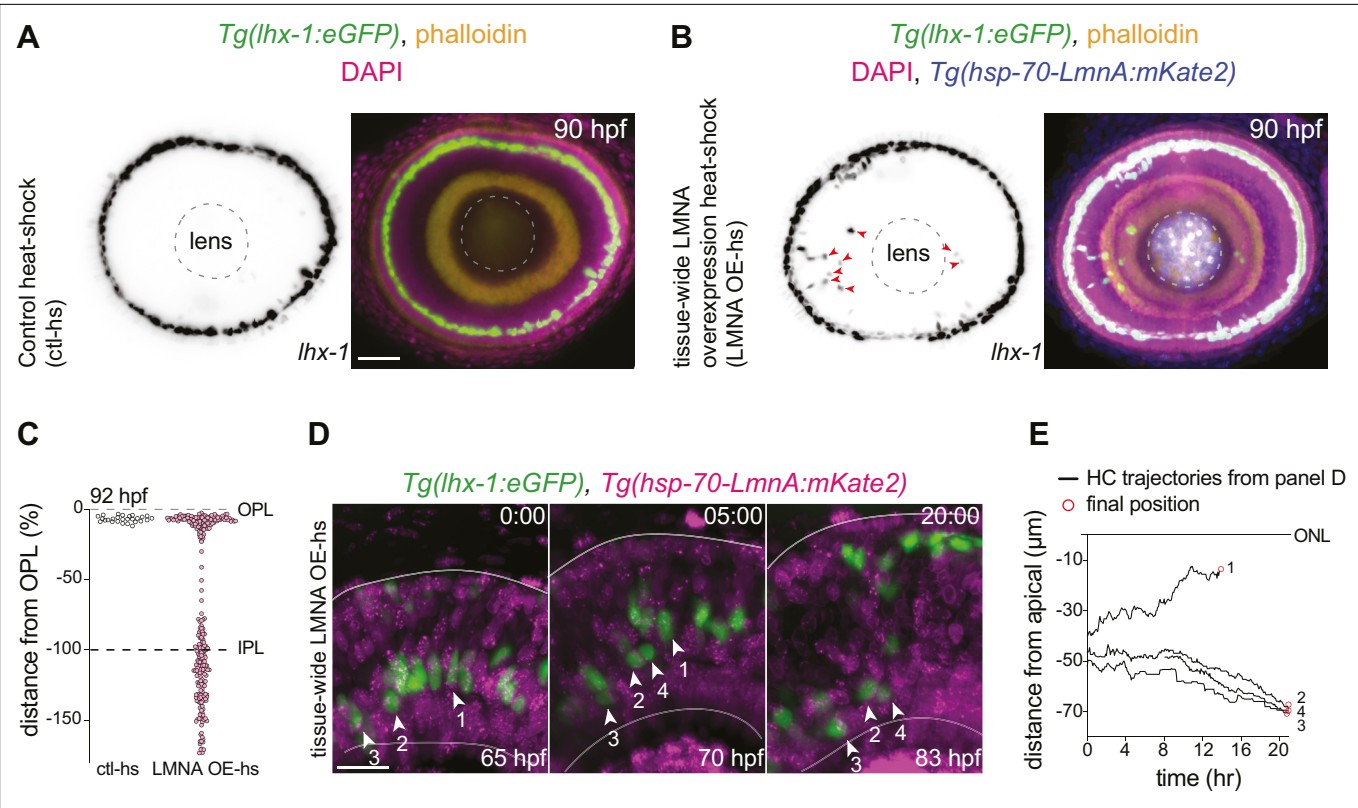

**Figure 5.** Interference with nuclear properties impairs efficient horizontal cell migration and layer formation. (**A–B**) Immunofluorescence images of (**A**) control heat-shocked (control-hs), and (**B**) LMNA overexpressing heat-shocked (LMNA OE-hs) at 90 hours post fertilization (hpf). *Tg(lhx-1:eGFP)* embryos were used for ctl-hs, and *Tg(lhx-1:eGFP)* × *Tg(hsp70:LMNA-mKate2)* double-transgene retinae for LMNA OE-hs experiments. *Tg(lhx-1:eGFP)* marks horizontal cells (HCs) (green), *Tg(hsp70:LMNA-mKate2)* is expressed in nuclear envelopes upon heat-shock (blue), DAPI labels nuclei (magenta), and phalloidin marks actin (yellow). Red arrowheads: ectopically positioned HCs, dashed circle: lens. Scale bar: 50 μm. (**C**) Position of HCs relative to outer plexiform layer (OPL) in control heat-shocked (ctl-hs) (n=27, N=5), and LMNA OE-hs (n=283, N=11) at 92 hpf. See *Figure 5—source data 1*. (**D**) Stills from a 20 hr time-lapse of *Tg(lhx-1:eGFP)* × *Tg(hsp70:LMNA-mKate2)* double-transgene retinae after heat-shock (LMNA OE-hs). Arrowheads: tracked HCs. Scale bar: 20 μm. (**E**) Migration trajectory of tracked HCs from (**D**). HC-2, HC-3, and HC-4 remained ectopically positioned in the basal part of the retina, while HC-1 successfully reached the HC layer. See *Figure 5—source data 1*. See also *Figure 5—figure supplement 1*.

The online version of this article includes the following source data and figure supplement(s) for figure 5:

**Source data 1.** *Figure 5C-E*.

**Figure supplement 1.** Analysis of Laminin overexpression in HC mosaic and whole tissue.

obstacle for migrating HCs. We postulated that this is most likely due to the intermingled axonal terminals of BCs, dendritic trees of ACs and RGCs, and MG processes which form a dense neuropil enriched with membrane (*Figure 2—figure supplement 1C-C''*).

To test this hypothesis, we set out to drive HCs to positions below the IPL before its formation, and then explore how IPL formation influenced HC migration. To this end, we genetically eliminated RGCs, using a validated *atoh7* morpholino (*Pittman et al., 2008*) that results in RGC depletion. In this condition, similar to *atoh7* mutants which completely lack RGCs (*Kay et al., 2004*; *Kay et al., 2001*), IPL formation is delayed and ACs are found at the most basal layer intermixed with occasional HCs (*Weber et al., 2014*).

Our immunofluorescence stainings of *Tg(lhx-1:eGFP)* retinae revealed that in *atoh7* morphants before emergence of the IPL at 50–55 hpf and, the maximal depth of HC migration increased significantly and that many HCs were located at the most basal side of the retina adjacent to the basement membrane (*Figure 6A* – bottom panel, red arrowhead, *Figure 6C*), a depth never seen for HCs in control embryos (*Figure 6A* – top panel, *Figure 6C*). Thus, upon interference with RGC emergence, HCs reached deeper positions in the tissue by moving beyond their typical basal stopping-point.

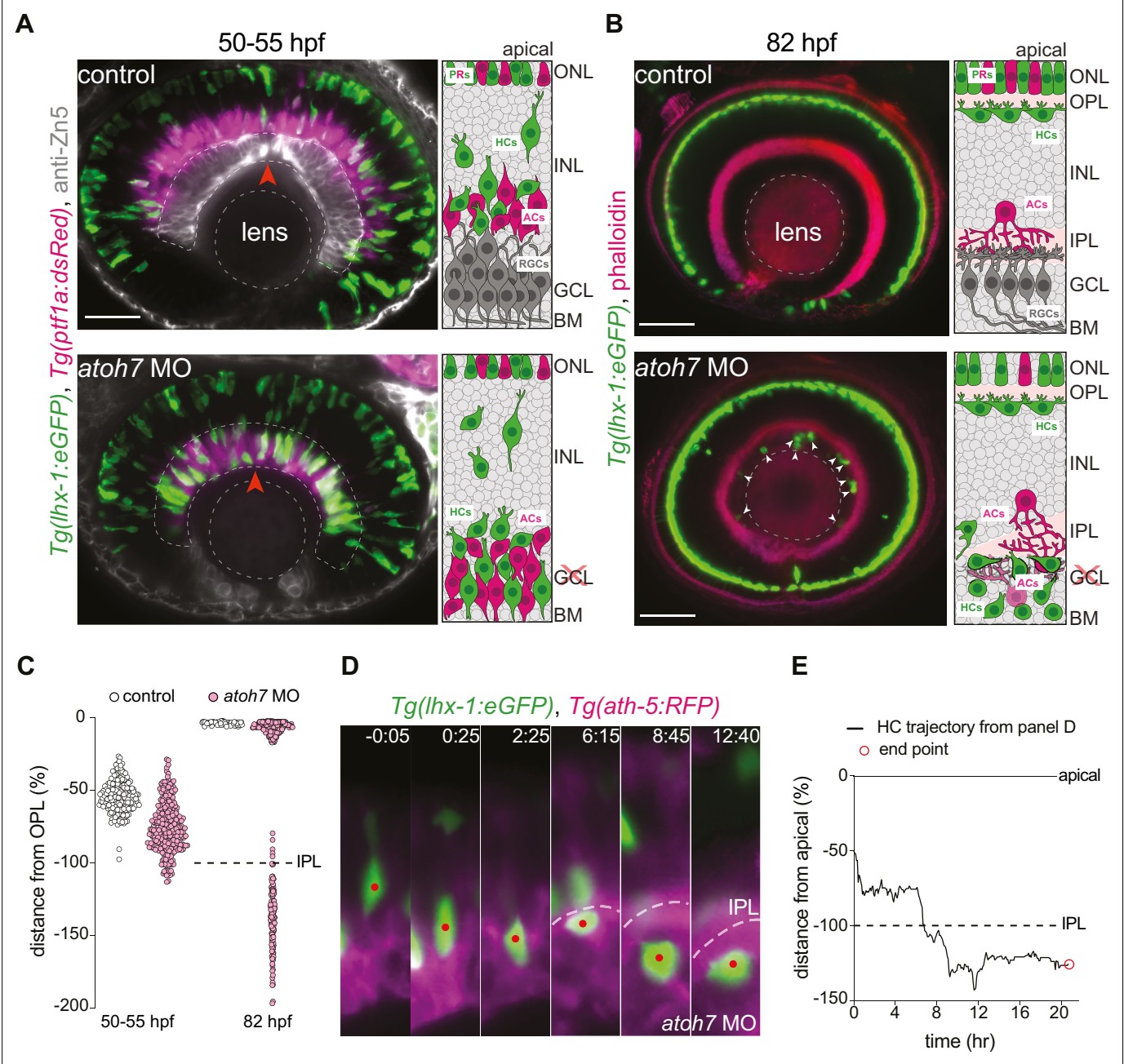

**Figure 6.** The inner plexiform layer (IPL) poses a barrier for horizontal cell apical migration. (**A–B**) Immunofluorescence images of control (top) and *atoh7* morpholino (*atoh7* MO) [bottom] at 50–55 hours post fertilization (hpf) (**A**), and at 82 hpf (**B**). Schemes of each condition are seen on the right. *Tg(lhx-1:eGFP)* is expressed in horizontal cells (HCs) (green), *Tg(ptf1a1:dsRed)* is expressed in amacrine cells (ACs) and HCs (magenta), anti-Zn5 marks retinal ganglion cells (RGCs) (gray), and phalloidin marks actin (magenta). In *atoh7* MO retinae, the most basal layer is devoid of RGCs (gray in control) and is instead filled with ACs (magenta), and HCs (green). (**A**) HCs reach ectopic basal positions adjacent to the basement membrane (BM) in *atoh7* MO retinae. Red arrowheads: BM. (**B**) Many HCs remain trapped beneath the IPL in *atoh7* MO retinae. White arrowheads: trapped HCs; dashed circle: lens. Scale bar: 50 µm. (**C**) Position of HCs relative to the outer plexiform layer (OPL) in control (n=196, N=12) and *atoh7* MO (n=261, N=14) retinae at 50–55 hpf, and control (n=76, N=11) and *atoh7* MO (n=312, N=15) at 82 hpf. At 50–55 hpf, the depth of HC migration increases in *atoh7* MO compared to controls. At 82 hpf, many HCs are ectopically located below the IPL in *atoh7* MO, while in controls all HCs reached their layer. See *Figure 6—source data 1*. (**D**) Time series of an HC (red dot) in *atoh7* MO retinae. Dashed line: the IPL. Scale bar: 20 µm. Time in h:min. (**E**) Migration trajectory of the tracked HC from (**D**). See *Figure 6—source data 1*. See also *Figure 6—figure supplement 1*.

The online version of this article includes the following source data and figure supplement(s) for figure 6:

**Source data 1.** *Figure 6C-E*; *Figure 6—figure supplement 1B-D*.

**Figure supplement 1.** Analysis of role of IPL formation on HC migration.

We next asked whether ectopically located HCs in *atoh7* morphant conditions were nevertheless able to reach their destination even when travelling from much more basal positions than what was seen in controls. To determine this, we monitored HC migration in control and *atoh7* morphant retinae until 82 hpf. We found that while all HCs reached the HC layer at 82 hpf in control retinae (*Figure 6B* – top panel, *Figure 6C*), many HCs were ectopically located at the most basal retinal layer in *atoh7* morphants (*Figure 6B* – bottom panel, arrowheads, *Figure 6C*). To understand why some HCs failed to reach the HC layer in *atoh7* morpholino conditions, we performed long-term time-lapse imaging from 48 to 90 hpf and monitored HC trajectories in *atoh7* morphants. We found that before IPL formation, HCs successfully turned back apically from their ectopic basal positions adjacent to the basal membrane (*Figure 6—figure supplement 1A* – dotted line) and reached the HC layer in *atoh7* morphants (*Figure 6—figure supplement 1A-B*). This implies that the ability of HCs to move back apically is not lost in the absence of RGCs but that during or after IPL formation, ectopically located HCs failed to pass through the IPL and hence remained trapped in ectopic locations beneath the IPL (*Figure 6D–E*, *Figure 6—figure supplement 1C-D*). Interestingly, these ectopically located HCs moved in all directions adjacent to the IPL (*Figure 6—figure supplement 1C-D*), until they underwent apoptosis, evidenced by their progressive fragmentations, immobility, and ultimately disappearance. Overall, we conclude that the IPL despite being more compressible than the nuclear layers (*Figure 2D–E"*) represents a barrier through which migrating HCs are not able to pass when trapped beneath. Thus, the IPL likely poses a limit to the morphological adaptability of HCs and their nuclei.

## Discussion

We here showed that HCs and their precursors follow principles of space adaptation strategies including repeated and reversible shape and direction change to navigate through the complex and densely packed environment of the developing retina. Because the migratory behavior and morphology of HCs share many hallmarks of amoeboid migration, we refer to HC migration as 'amoeboid-like migration'. To the best of our knowledge, this is the first study describing a cell ultimately becoming a neuron that undergoes amoeboid-like migration in a part of the developing CNS. We further uncovered that changing tissue properties can feedback on the efficiency of amoeboid-like migration and layer formation, most likely by influencing the space negotiation capability of HCs.

As opposed to neocortical neurons including projection neurons which en route to their destination switch from multipolar to bipolar morphology and resume unidirectional radial migration along radial glia fibers (*Nadarajah et al., 2003*; *Nadarajah et al., 2001*; *Cooper, 2014*), HCs do not travel along radially oriented progenitors (MGs), or neurons (BCs). It is however possible that HCs use their neighboring neurons as substrate over which to migrate in all directions (x, y, z) to reach their destination.

During long stretches of their migration, HCs follow unconventional tortuous migratory tracks while frequently alternating between radial (up and down) and tangential (lateral) routes. Such migration trajectories also set HCs apart from other emerging retinal neurons including PRs (*Rocha-Martins et al., 2021*) and RGCs (*Icha et al., 2016a*) that display bipolar morphologies and remain constrained to radial migratory routes. While these neurons also undergo some deformations in the crowded retina, their migration success most likely does not depend on these deformations as they remain anchored to the tissue surface throughout their migration cycle. In contrast, HCs migrate to their destination with no attachments. We hence speculate that the flexible trajectories (*Amini et al., 2019*) and space adaptation strategies allow HCs to move in all dimensions to overcome obstacles within the crowded environment. Thus, while HCs reproducibly and robustly reach the HC layer, their path selection is not intrinsically programmed as opposed to most other retinal neurons (e.g., PRs and RGCs) but is rather influenced by the cellular surroundings and the tissue-scale parameters they encounter in their local environment.

The amoeboid-like migration mode exhibited by HCs is not based on bleb formation but correlates with multiple highly dynamic actin-filled protrusions. The exact nature of these protrusions, and whether they directly drive HC migration, explore potential environmental cues or both, remain to be further investigated. Our observation that protrusions are simultaneously extended toward multiple directions, especially during periods in which HCs stay stationary, suggests that they are rather involved in probing the tissue environment and pathfinding than directly propelling the movement. Such exploratory roles have been proposed in amoeboid moving cells including *Dictyostelium* during chemotaxis, leukocytes, and neutrophils (*Gupton et al., 2005*; *Wu et al., 2012*; *Leithner et al., 2016*;

*Vargas et al., 2016*; *Fritz-Laylin et al., 2017*; *Gerisch and Hess, 1974*). Future experiments that specifically interfere with protrusion formation or maintenance will shed light on their exact role in HC migration and layer formation.

We currently do not understand the mechanism(s) and forces that move HCs forward in the densely packed developing retina. Many amoeboid migrating cells display a front-rear polarity with asymmetric enrichment of stable F-actin filaments at the highly contractile uropod (*Hind et al., 2016*; *Bergert et al., 2012*; *Lammermann and Sixt, 2009*). Our finding that migrating HCs display strikingly similar polarized morphology implies that they may also use uropod contraction as a pushing force to move forward. Unraveling the spatiotemporal molecular machineries of cell polarity, force generation, the signaling and the cytoskeletal elements that drive them will be exciting areas for future studies.

Using in vivo Brillouin microscopy, we showed that the Brillouin shift maps of the INL remain relatively homogenous throughout HC migration. That no obvious compressibility gradient was observed along the apico-basal axis of the INL during HC migration implies that tissue compressibility could have a permissive rather than an instructive role for HC migration. The fact that interfering with tissue-wide components such as properties of the nuclear lamina of both HCs and their surrounding cells impedes HC migration efficiency and successful layer formation further argues in this direction.

An additional tissue-wide feature that influences HC migration is the emergence of the IPL. We previously reported that the depth of HC migration correlates with IPL emergence and that once it is formed, HCs do not pass beyond it on their apical-to-basal journey (*Amini et al., 2019*). This together with our finding that HCs get trapped beneath the IPL in RGC-depleted retinae strongly suggests that the IPL acts as a steric hindrance through which HCs cannot penetrate in either direction. It is plausible that the fibrillar arrangement of axonal and dendritic processes of the IPL poses a net-like obstacle with low porosity that is below the deformation capability threshold of HC nuclei, which mechanically traps HCs despite their ability to move. This idea is in line with studies that showed that migration efficiency is optimal at pore diameters that match or range slightly below the diameter of cell's nucleus. Hence, while we cannot exclude that chemical signals also contribute to entrapment of HCs below the IPL, our result that HCs do not reorient upon hitting the mature IPL in *atoh7* morphants argues that the IPL does not act as a chemical repulsive barrier but most likely poses a structural constraint through which HCs fail to pass. In future, studying the mechanochemical composition of the IPL will provide insights on whether, how, and to what extent it imposes a physical and/or chemical restriction on HC migration.

Together, our data points toward the idea that the space adaptation migration mode carried out by HCs allows them to follow the 'path of least resistance' (*Weber et al., 2013*; *Renkawitz et al., 2019*; *Kameritsch and Renkawitz, 2020*) when moving in a cell-dense tissue environment in the retina that does not seem to provide substantial matrix components. While we so far did not find evidence that protease-dependent strategies were used by HCs to move through the tissue, we currently cannot exclude that they are involved at some stage to make migration more efficient. Another factor we cannot yet exclude is that proteolytic degradation of cell-cell junctions helps HCs to move through the crowded tissue. Thus, how exactly the 'path of least resistance' is found by HCs, which additional factors might help finding it and whether proteolytic-dependent remodeling schemes aid HCs in moving forward, will be exciting avenues for future studies.

This will entail to find the external cues that guide HC migration and ensure that HCs always find their accurate functional position while avoiding entrapment within the crowded retina. Such cues could either come in the form of mechanical gradients or chemical signaling or a combination of both. It will be important to find the source and nature (e.g., attractive vs. repulsive) of these cues, to understand how they change in space and time, and how HCs sense, integrate, and prioritize these cues from the local structural and emerging geometrical features of their surroundings.

Taken together, this study reveals that in addition to the numerous migration modes previously characterized in the developing nervous system, cells can also undergo amoeboid-like migration in an important part of the developing CNS, the retina. We believe it likely that the ability to undergo direction, cell- and nuclear-shape changes allows HCs to evade rather than degrade encountered barriers in the crowded tissue environment. It was shown recently that axon growth of CNS neurons also exhibits amoeboid principles (*Santos et al., 2020*). Further, as multipolar migration modes are observed in many parts of the developing CNS, it is likely that amoeboid-like migration and behavior is widespread in diverse systems beyond the retina. Similarities and differences of what influences

such amoeboid-like migration in different systems will teach us more about the intricate development of our brain.

# Materials and methods
## Zebrafish work
### Resource availability
All transgenes and reagents generated in this study are available from the Lead Contact without restriction. Further information and requests for resources and reagents should be directed to and will be fulfilled by the Lead Contact, Caren Norden (cnorden@igc.gulbenkian.pt).

### Zebrafish husbandry
Wild-type TL zebrafish (*Danio rerio*) and transgenic lines were maintained and bred at 26°C as previously described. Embryos were raised at 28.5°C or 32°C and staged in hpf according to *Kimmel et al., 1995*. Embryos were kept in E3 medium, which was renewed daily and supplemented with 0.2 mM 1-phenyl-2-thiourea (PTU) (Sigma-Aldrich) from 8±1 hpf onward to prevent pigmentation. All animal work was performed in accordance with the European Union (EU) directive 2010/63/EU, as well as the German Animal Welfare Act.

### Zebrafish transgenesis
To generate *Tg(hsp70:LMNA-mKate2)*, a stable transgenic line containing heat-shock-inducible LMNA 1 nl of hsp70:LMNA-mKate2 (*Yanakieva et al., 2019*) was injected at 36 ng/μl, together with Tol2 transposase RNA at 80 ng/μl in double-distilled (dd)$H_2O$ supplemented with 0.05% phenol red (to visualize the injection material) (Sigma-Aldrich) into the cytoplasm of one-cell stage wild-type embryos. $F_0$ embryos were raised until adulthood. Germline carriers displaying mKate signal upon heat-shock treatment at 37°C, for 20 min, at 24 hpf were identified in $F_0$ progeny. Carriers were then outcrossed with wild-type fish.

### Transgenic lines
Refer to *Supplementary file 1*a for a list of transgene lines.

### DNA injections
To mosaically label HCs or express proteins of interest in the zebrafish retina, DNA constructs were injected into the cytoplasm of one-cell stage embryos. Constructs were diluted in dd$H_2O$ supplemented with 0.05% Phenol Red (Sigma-Aldrich). Injected volumes ranged from 1 to 1.5 nl. DNA concentrations were 20–30 ng/μl and did not exceed 45 ng/μl when multiple constructs were injected. See *Supplementary file 1b* for a list of injected constructs.

### Morpholino experiments
To inhibit RGC formation, *atoh7* morpholino (5'-TTCATGGCTCTTCAAAAAAGTCTCC-3') (Gene Tools) was injected at 2 ng per embryo into the yolk of one-cell embryos. *p53* morpholino (5'- GCGCCATT GCTTTGCAAGAATTG-3') (Gene Tools) was co-injected at 2–4 ng per embryo to reduce toxicity and cell death.

### Heat-shock
To induce expression of heat-shock promoter (hsp70)-driven DNA constructs and transgenes (except *Tg(hsp70:LMNA-mKate2)*), the Petri dish with 36–42 hpf embryos was placed into a water bath set to 37°C for 30 min or 39°C for 15–20 min. Imaging was started 3–6 hr after heat-shock. For hsp70-driven LMNA overexpression, *Tg(hsp70:LMNA-mKate2) and hsp70:LMNA-mKate2 injected embryos were transferred to E3-containing 15 ml tubes which were preheated* in a water bath for 30 min. Heat-shock started 3–10 hr before imaging at 45–48 hpf for 30 min at 39°C, or at 50–60 hpf for 15 min at 42°C. Embryos which displayed higher mKate fluorescence intensity after heat-shock but did not display developmental delays or defects were picked for experiments.

## Whole-mount staining of zebrafish embryos

Zebrafish were manually dechorionated and fixed overnight in 4% paraformaldehyde (PFA) (Sigma-Aldrich) in PBS at 4°C. Embryos were washed five times for 15 min with PBS-Triton (PBS-T) 0.8%. Embryos were then permeabilized with 1× Trypsin-EDTA in PBS on ice for different time periods depending on the developmental stage (12 min for 42–50 hpf, 15 min for 56–60 hpf, 20 min for 70–90 hpf). The permeabilization solution was replaced with 0.8% PBS-T and embryos were kept for an additional 30 min on ice before rinsing twice with 0.8% PBS-T. Embryos were then blocked in 10% goat or donkey serum (blocking serum) in 0.8% PBS-T for 3 hr at room temperature and were subsequently incubated with primary antibodies diluted in 1% blocking serum in 0.8% PBS-T for 3 days at 4°C. They were then washed five times for 30 min with 0.8% PBS-T. After blocking, embryos were incubated with appropriate secondary antibodies and DAPI in 1% blocking serum in 0.8% PBS-T for 3 days at 4°C. Embryos were washed four times for 15 min with 0.8% PBS-T before storage in PBS at 4°C until imaging. Refer to *Supplementary file 1c* for antibodies used.

## Blastomere transplantations

Transplantation dishes were prepared by floating a plastic template in a Petri dish that was half-filled with 1% low-melting-point agarose in E3. Once the agarose solidified, plastic templates were gently removed, leaving an agar mould that contained rows of wells to hold embryos. Embryos at stages high to sphere were dechorionated in pronase (Roche) and dissolved in Danieu's buffer. Dechorionated embryos were transferred to wells in agarose molds using a wide-bore fire-polished glass pipet. Approximately at the 1000-cell stage, cells from the donor embryos were transplanted into the animal pole of the acceptor embryos using a Hamilton syringe. Transplanted embryos were kept on agarose for about 3–5 hr and then transferred onto glass dishes that contained E3 medium supplemented with 0.003% PTU and antibiotics (100 U of penicillin and streptomycin, Thermo Fisher Scientific). Transplanted embryos were identified via fluorescence and imaged from 42 hpf for 24–30 hr.

## Image acquisition

### In vivo light sheet fluorescent imaging

Imaging was performed on a Zeiss Light sheet Z.1 microscope as previously described (*Icha et al., 2016b*). The system was operated by the ZEN 2014 software (black edition). Briefly, embryos were manually dechorionated and mounted in glass capillaries in 0.9% low-melting-point agarose (in E3) supplemented with 240 μg/ml of tricane methanesulfonate (MS-222; Sigma-Aldrich). The sample chamber was filled with E3 medium containing 0.01% MS-222 and 0.2 mM PTU (Sigma-Aldrich) and was kept at 28.5°C. Z-stacks spanning the entire width of the retinal neuroepithelium (90–100 μm, depending on the developmental stage) were recorded with 1 μm optical sectioning every 5 min for 10–40 hr with a Zeiss Plan-Apochromat 20× water-dipping objective (Carl Zeiss Microscopy; NA 1.0) and two Edge 5.5 sCMOS cameras (PCO), using double-sided illumination mode. Z-stacks were recorded every 5 min for 24–40 hr, with double-sided illumination mode. For protrusion monitoring experiments (*Figure 4A–C"*, *Video 9*) images were taken every 1 min for 10–14 hr (Section Protrusion tracking and analysis). Imaging started between 42 and 48 hpf, except for LMNA overexpression which started at 48–53 hpf.

### Confocal scans

Fixed samples were imaged in a laser-scanning microscope (LSM 700 inverted, LSM 880 Airy upright; ZEISS) or point scanning microscope (two-photon inverted; ZEISS) using the 40×/1.2 C-Apochromat water immersion objective (ZEISS). The samples were mounted in 1% agarose in glass-bottom dishes (MatTek Corporation) filled with E3 medium and imaged at room temperature. The microscopes were operated with the ZEN 2011 (black edition) software (ZEISS).

## Image processing and analysis

### Sample drift correction

First, maximum projected sub-stacks (five z slices) of the raw live images were generated in Fiji. XY-drift of 2D stacks was then corrected using a manual drift correction Fiji plug-in created by Benoit Lombardot (Scientific Computing Facility, Max Planck Institute of Molecular Cell Biology and Genetics,

Dresden, Germany). The script can be found on imagej.net/Manual_drift_correction_plugin. The drift-corrected movies were then used for tracking migrating HCs.

### Tracking migrating HCs

The migrating HCs were manually tracked by following the center of the cell body in 2D drift-corrected images using MTrackJ plug-in in Fiji (*Meijering et al., 2012*).

### Deconvolution

The raw LSFM data was deconvolved in ZEN 2014 software (black edition, release version 9.0) using the Nearest Neighbor algorithm. Minimal image pre-processing was implemented prior to image analysis, using open-source ImageJ/Fiji software (fiji.sc). Processing consisted of extracting image subsets or maximum intensity projections of a few slices. Processed files were analyzed in Fiji.

### Morphodynamic analysis of HC migration

To quantify HC cell and nuclear morphodynamics, the free and open-source platform for bioimage analysis Icy (*de Chaumont et al., 2012*) (http://icy.bioimageanalysis.org) was used to automatically digitize cell and nuclear contours. The 'Active Contours' plug-in was used to segment the contours of HC cell and nuclear outlines during migration and in mitosis (Materials and methods). Because the retina is densely packed, the segmentation only worked when the cell of interest was singled out from the background and had enough distance from neighboring cells. To meet this goal, we used two different approaches: one-cell transplantation (see Sections Heat-shock, Blastomere transplantations) and 2-DNA injection (see Section 1Zebrafish transgenesis). A step-to-step manual of the protocols and plug-ins to measure cell and nucleus morphodynamics is available in (http://icy.bioimageanalysis.org) (*Manich et al., 2020*).

In this study, we extracted the following shape descriptors from Icy analysis: perimeter (μm), sphericity (%), elongation ratio (a.u.). 'Perimeter' measures the perimeter of the region of interest (ROI) in micrometers. 'Sphericity' is a measure of how similar to a sphere the ROI is. 'Elongation ratio' is a scale factor given by the ratio between the first and second ellipse diameters of an ROI. The minimum value is 1 (for a round object).

### Protrusion tracking and analysis

Protrusions were analyzed using light-sheet time-lapse video recording of *Tg(ptf1a:Gal4-VP16, UAS:gap-YFP) at 1 min intervals*. The protrusion tips were manually tracked simultaneously with HC centroids using the MTrackJ plug-in in Fiji (Section Tracking migrating HCs). The angle between the protrusion and the direction of HC movement was defined as the angle between the unit vector defined by the direction of HC movement and the unit vector pointing to the protrusion tip from the HC centroid. All plots for this part of the analysis were created in Matplotlib (*Hunter, 2007*).

### Utrophin fluorescence intensity distribution profiles

Utrophin fluorescence intensity distribution profiles of migrating HCs were measured in Fiji by drawing a line (width = 3) along the cell axis at each time point. The utrophin signal intensity was measured using the max projection of three consecutive central z planes of the cell.

## In vivo Brillouin light scatter microscopy

### Mounting of zebrafish larvae for in vivo Brillouin

Embryos were anesthetized in MS-222 (0.02% in E3; Sigma-Aldrich) for approximately 20 min and placed in a lateral position on a glass-bottom dish suitable for optical imaging. Some specimens were placed on a polyacrylamide gel that acted as a spacer between the glass bottom and the embryo (for gel preparation, see *Schlüßler et al., 2018*). A drop (200 μl) of low-gelling-point agarose (1% in E3, 30°C; Sigma-Aldrich) was used to immobilize the embryo. Immobilized larvae were then immersed in MS-222 (0.02%) and PTU (0.003%, Sigma-Aldrich) containing E3 during imaging. All embryos were released from the agarose embedding between Brillouin measurements and kept under standard conditions as described in Section Resource availability.

## Brillouin microscopy setup and data analysis

The Brillouin shift measurements were performed using a custom-built Brillouin microscopy setup described in Schlüßler, Möllmert et al. 2018. The setup consists of a frequency-stabilized diode laser with a wavelength of 780.24 nm (DLC TA PRO 780; Toptica, Gräfelfing, Germany), a confocal unit employing a Plan Neofluar 20× objective (Carl Zeiss Microscopy; NA 0.5) and a two-stage virtually imaged phased array spectrometer. The setup was controlled using a custom program written in C++ (https://github.com/BrillouinMicroscopy/BrillouinAcquisition; *GuckLab – Brillouin-Microscopy, 2022*; copy archived at swh:1:rev:4321f1e403b644d01e5ef0e743a437cee161bbd2). The improvement of the Brillouin microscopy setup by adding a Fabry-Pérot interferometer lead to the suppression of previously described strong reflections from the glass surface and rendered the gel spacer in subsequent measurements unnecessary. The data acquired was evaluated using a custom MATLAB (The MathWorks, Natick, MA) program (https://github.com/BrillouinMicroscopy/BrillouinEvaluation; copy archived at swh:1:rev:3070654dc2b03fcacef8bd236935105cab128964; *Schlüßler, 2022*).

## Acknowledgements

We thank Otger Campàs, Rita Mateus, Teije Middelkoop, Carl Modes, Jaakko Lehtimäki, and Jacqueline Tabler for constructive feedback on the manuscript. We are grateful to Heike Hollak and Sylvia Kaufmann for their technical support. We thank the Light Microscopy Facility of the Max Planck Institute of Molecular Cell Biology and Genetics (MPI-CBG), especially Jan Peychl, for their advice and technical support on image acquisition. We are also grateful to the Fish Facility of the MPI-CBG for technical assistance. We thank Gayathri Nadar from the Scientific Computing Facility at the MPI-CBG for discussions on data analysis.

RA thanks Stephan W Grill and Otger Campàs for financial support and Rita Mateus for generously providing lab space and reagents for experiments. RA is grateful to Ivan Baines and Anthony Hyman for their help, advice and support. RS thanks Simon Alberti for financial support.

R.A. was a research fellow of the Natural Sciences and Engineering Research Council of Canada (NSERC) from 2017-2019 and the Fonds de la Recherche du Québec-Santé (FRQ-S) from 2019-2020. C.N. was supported by MPI-CBG, the FCG-IGC, Fundação para a Ciência e a Tecnologia Investigator grant (CEECIND/03268/2018) and an ERC consolidator grant (H2020 ERC-2018-CoG-81904).

## Additional information

### Funding

| Funder | Grant reference number | Author |
|---|---|---|
| H2020 European Research Council | ERC-2018-CoG-81904 | Caren Norden |
| Natural Sciences and Engineering Research Council of Canada | 502961 | Rana Amini |
| Fonds de Recherche du Québec - Santé | 35510 | Rana Amini |
| Max Planck Institute of Molecular Cell Biology and Genetics | open access funding | Rana Amini |
| Fundação para a Ciência e a Tecnologia Investigator grant | CEECIND/03268/2018 | Caren Norden |

The funders had no role in study design, data collection and interpretation, or the decision to submit the work for publication.

## Author contributions
Rana Amini, Conceptualization, Data curation, Formal analysis, Investigation, Methodology, Visualization, Writing - original draft, Writing - review and editing; Archit Bhatnagar, Investigation, Methodology, Validation, Writing - review and editing; Raimund Schlüßler, Investigation, Methodology; Stephanie Möllmert, Jochen Guck, Methodology; Caren Norden, Conceptualization, Data curation, Funding acquisition, Project administration, Supervision, Validation, Writing - original draft, Writing - review and editing

## Author ORCIDs
Rana Amini ⬥ http://orcid.org/0000-0002-3974-5072
Raimund Schlüßler ⬥ http://orcid.org/0000-0003-3752-2382
Caren Norden ⬥ http://orcid.org/0000-0001-8835-1451

## Ethics
All animal work was performed in accordance with the European Union (EU) directive 2010/63/EU, as well as the German Animal Welfare act.

## Decision letter and Author response
Decision letter https://doi.org/10.7554/eLife.76408.sa1
Author response https://doi.org/10.7554/eLife.76408.sa2

---

# Additional files

## Supplementary files
• MDAR checklist

• Supplementary file 1. Tables of: Transgenic lines; DNA constructs; Antibodies; and relevant links to Figures.

## Data availability
All data generated or analysed during this study are included in the manuscript and supporting file; Source Data files have been provided.

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
