## [Editor Report]

The authors probe the role of multipolar migration of horizontal cells in the zebrafish retina. The results reveal amoeboid-like migration enabling cell movements to adapt to environmental spatial constraints in the crowded retina including cell and nuclear shape changes and rearward polarization of stable F-actin.

---

## [Decision Letter]

[Editors' note: this paper was reviewed by Review Commons.]

---

## [Author Response]

Reviewer #1 (Evidence, reproducibility and clarity (Required)):This work by Amini and co-authors investigate how an interneuron class in the vertebrate retina, horizontal cells (HCs), that reside in the inner nuclear layer, migrate to their final location. The authors capitalize on the advantages of the zebrafish retina to track the trajectories of HCs by live time-lapse imaging. The authors conclude that HCs don't undergo somal translocation or cell-guided migration as described for pyramidal neurons in the neocortex but rather use amoeboid-like migration to navigate to their final destination. Using an array of genetic tools this study reveals the cell biology behind HC migration.I have no major concerns – this is a well-written and well-illustrated paper. The authors are careful not to overstate their findings. There are some minor issues which I address below.1) One thing that was confusing to me was that the authors show a diagrammatic representation in Figure 1B of a mitotic division occurring at the apical surface, with one daughter proceeding to a basal location at the interface of the INL and IPL before moving apically towards the outer INL near the OPL. My understanding of previous work from this lab (Weber et al. 2014; Amini et al. 2019) is that HCs are generated by terminal divisions that occur within the INL. Wouldn't that mean that the cells that the authors describe moving from the basal parts of the INL to more apical locations are actually HC progenitors and not post-mitotic HCs?

This is a good point made by the reviewer that we addressed by text changes in the introduction of the updated manuscript. We now clarify that no differences have been found between HCs and their precursors in migration profile, which is why we refer to both cell types as ‘HC’ in the remainder of the manuscript.

We now also clarified the schematic in Figure 1B to avoid confusion adding a legend for all cell types.

We also changed the title of the manuscript to:

“Ameboid-like migration ensures correct horizontals cell layer formation in the developing vertebrate retina”, to avoid confusion as not all cells followed were already fully differentiated neurons.

2) Somewhat perplexingly, the caption to Figure 3B reads: " Schematic representation of a typical basal-to-apical migration trajectory of an HC which undergoes mitosis en route. During mitosis, HCs switch from an elongated shape into a spherical morphology." Since HCs are post-mitotic it is not HCs but rather the progenitors of HCs that undergo mitosis en route.

See our point above. We now changed the manuscript, schematic and title accordingly to do justice to the fact that we follow HC precursors as well as maturing HCs. The figure legend also has been amended.

3) It would be worthwhile explaining why different transgenic lines are used to track HCs – ptf1a, lhx1, and sometimes even trβ2 (isn't the latter a red cone marker?)

We added information in Supplementary File 1(please refer to supplementary file 1a, 1b, 1c, 1d). Regarding trβ2, the reviewer is correct. However, trβ2 was also shown to label emerging committed HC precursors and HCs (Suzuki et al., 2013 and Amini et al., 2019).

Are there advantages to using specific lines for particular experiments?

Briefly, we used different transgenic lines or DNA plasmids depending on the question asked. For example, to monitor protrusion activity of HCs, we used *Tg(ptf1a:GAL4-VP-16, UAS:gap-YFP)* which labels membrane of committed HC precursors and HCs. In case overall cell behavior was followed, cytosolic HC markers (e.g. trβ2, lhx-1) were used. We would like to point out that we previously showed that migration and mitotic behavior of committed HC precursors and HCs labeled by these different markers do not differ (Amini et al., 2019).

4) Given that the retina is a dense tissue could the authors comment as to whether all migrating cells undergo cellular deformations to reach their final layer; or whether this is only a feature of HCs they describe in this paper.

This is a good point and we added some additional explanations in the discussion to better clarify it.

Briefly, other cell types also undergo some shape changes. However, their deformations are less pronounced as they undergo less complicated migration modes compared to HCs. For example, retinal ganglion cells and photoreceptors both keep their apical attachment during migration (Icha et al., 2016, Rocha et al., 2021). Thus, these cells remain constrained to radial routes and do not need to deform as much during migration as they move in one dimension. Bipolar cells are seen to reside more or less where they divide and do not undergo massive translocation post mitosis (Weber et al., 2014, Engerer et al., 2017). Last but not least, ACs only undergo a very short multipolar migration phase after losing their apical process and reside in the amacrine cell layer shortly thereafter.

In contrast to all these examples, HCs and their precursors lack attachments throughout most of their journey and thus need to deform substantially and move in all dimensions (x, y, z) to reach their destination. This makes HCs and their precursors the only cell-type in the retina that depends on deformation and pathfinding strategies for proper migration.

5) On page 18 the authors refer to (Kay et al. 2004, Kay et al. 2001, Pittman, Law and Chien 2008) as refs using a "atoh7 morpholino". Only the Pittman et al. used a morpholino – the other 2 papers seem to have used a mutant to prevent RGC generation.

We thank the reviewer and have addressed this point in our referencing.

6) In Supplementary Figure 1D why don't we see green HCs at the outer part of the INL, even at the last timepoint shown?

Supplementary-Figure 1D (now: Figure 1—figure supplement 1D) shows an embryo at 60 hpf which is the peak of apical migration of HCs and their precursors. At this stage, the majority of HCs are seen in the mid-INL on their way to apical-INL.

Full lamination is expected at later stages (around 70 hpf) but our live-imaging for this experiment ended before this point. The take home message of this Figure, together with Figure 1C and D was that HCs migrate independently of BC presence, early and late. Thus, full HC lamination was not the main focus of this experiment but can be well appreciated for example in Figure 1E at 70 and 90 hpf, and Figure 5A at 90hpf.

7) Could the authors comment on why HC migration following induced over-expression of hsp-70-LmnA:mKate2 did not have a more drastic effect than shown in Figure 5B.

We tried different heat shock treatment conditions to find the optimal conditions at which development of the embryo and the retina were not stalled upon heat shock neither in control nor in *Tg(hsp-70-LmnA:mKate2)* animals. To this end, we chose a medium heat shock-induced overexpression regime (explained in Materials and methods) as stronger heat-shock resulted in embryonic lethality or drastic developmental abnormalities in the retina and/or other tissues in both control and *Tg(hsp-70-LmnA:mKate2)* animals. In addition, as development proceeds the efficacy of heat shock-induced LMNA-overexpression wanes, explaining why the effects of LMNA overexpression are not more drastic.

8) Some grammatical errors: For example on page 5 – "However, if, how, and to what extend cellular and tissuewide properties influence HC movements towards layer formation remained unexplored". Should be: to what extent.

We carefully scanned the manuscript and corrected these mistakes.

Articles are sometimes lacking – particularly before INL, IPL – 2 examples: page 10 "…..space was detected in between the neighboring nuclei in INL (Sup-Figure 2B-B')". Page 19 "Because these basal locations are completely different from INL wherein HC migration occurs,…."

We carefully scanned the manuscript and added articles where necessary.

Reviewer #2 (Evidence, reproducibility and clarity (Required)):Summary:Provide a short summary of the findings and key conclusions (including methodology and model system(s) where appropriate).Amini et al. extended their high-quality imaging-based analysis in the developing zebrafish retina focusing on the horizontal cell (HC) lineage to newly address through which cell-intrinsic and tissue-level mechanical mechanisms the second half of HC migration (i.e., post-reversal, apical-ward phase by the multipolar-shaped HCs [MHCs]) proceeds. The authors show that MHC migration is non-radial, and that it takes places in a denselypacked tissue environment, with cellular and nuclear deformations, front-back polarization, and variable centrosomal position changes. The authors categorize this MHC migration mode as "amoeboid", which has not yet been extensively described in the developing CNS. As an experiment to functionally manipulate a cell-intrinsic factor, the nuclear lamina molecule Lamin A, which is usually only weakly expressed in the developing zebrafish retina, was over-expressed in HCs thereby reducing their nuclear deformability, and the authors found that many LaminA-high MHCs could not go apically, suggesting a model that nuclear deformability under a crowded environment is important for amoeboid MHC migration. To ask a possible barrier-like role (i.e., pushing from the rear of MHCs) by the inner plexiform layer (IPL), the authors depleted retinal ganglion cells (RGCs) which normally occupy a basal-most zone of the developing retinal wall, and found that MHCs could not take a normal apical position, with many dislocated to or left in an RGC-free zone near the basal/inner retinal surface.Major comments:1) Multipolar migration is shown by HCs that are still capable of division ("HCpr" [progenitor or precursor], as stated in Development 2019 by the same authors, is included), thus cells displaying amoeboid migration should not simply/entirely be called "neurons". Thus, title, which is currently incorrect, should carefully be revised. This is not a story of "neuronal migration", HC migration. Saying that HCs divide or that some of them are mitotic is acceptable.

This is a good point made by the reviewer that we addressed by text changes in the introduction of the updated manuscript. We now clarify that no differences have been found between HCs and their precursors in migration profile, which is why we refer to both cell types as ‘HC’ in the remainder of the manuscript.

We now also clarified the schematic in Figure 1B to avoid confusion adding a legend for all cell types.

We also changed the title of the manuscript to:

‘Ameboid-like migration ensures correct horizontals cell layer formation in the developing vertebrate retina’, to avoid confusion as not all cells followed were already fully differentiated neurons.

2) As I intentionally use "MHC" in this comment (though not necessarily requesting the authors to use this), the authors should be careful not to confuse the readers as to whether this study asks all the HC migration phases (entire U-shaped route) from the apical detachment to the final positioning (via reversal in between), or it instead focuses on the second (post-reversal, apical-ward, multipolar) half only (I think so). Figure 1B is vague about what the question is. Whether the initial apical-to-basal, bipolar phase is also amoeboid or not should be clearly stated in the text.

We now clarified this point throughout the manuscript and in the title. We also clarified the schematic in Figure 1B and added legends to make it more intuitive which cells are referred to where. The schematic shows a typical trajectory of an HC progenitor and its committed precursors to orient the reader.

3) Okuda et al. (J Biomechanics 46, 1705, 2013) provided a vertex model using an apically-concave (and basally convex) model epithelial wall system, showing that constriction/narrowing of the apical surface and the local cell number (i.e., tissue volume) increase together generate a basal-ward intramural pushing force. I think this model may be applicable to an apically convex and basally concave retinal wall. Even though the concave retinal basal surface may not have a strong constriction/contraction (as the model concave apical surface exhibits via actomyosin), its boundary effect and nearby occupancy by RGCs (and zonal volume increase by maturation/differentiation of individual RGCs) might generate a local crowding/compression as well as a basal-toapical pushing force. Experimental loss of RGCs in the basal-most layer, not only IPL, in the present study might have reduced such a hypothetical peri-basal crowding and an associated basal-to-apical intra-retinal force, as in the mathematically simulated apical surface (Okuda et al., 2013). Reduced apical-ward displacement of amoeboid-like migrating MHCs might be best explained if what this RGCs-ablation experimentally generated was taken as not only removal of IPL but also peri-basal decompression. Therefore, this RGC-ablation experiment might have much greater significance than just modestly telling that IPL plays a barrier role behind. It might be showing us a major driving force that the developing retina generates to physically drive MHC amoeboid migration using its inherent geometry and pre-existing cellular placement/packaging.

We thank the reviewer for this thorough explanation and comment about possible further implications of our study. We noted in the discussion that additionally to the steric hindrance also other emerging mechanical factors including tissue architecture are possibly playing a role.

Regarding the vertex model (Okuda et al., 2013), we think that it is unlikely that “RGCs generate a local crowding/compression as well as a basal-to-apical pushing force” as the reviewer suggested for the following reasons:

1) We would like to point out that in RGC-depleted retinae, the IPL is formed, however with a delay compared to control retinae. While before IPL formation, HCs and their precursors successfully move out of the most-basal layer towards their final position (Figure 6—figure supplement 1A-B), they keep residing beneath the IPL after its formation (Figure 6—figure supplement 1C-D). This together with our previous study where we showed that HCs cannot basally bypass the IPL (Amini et al., 2019), suggests that the IPL functions as a barrier for migrating HCs. We provided additional images in the supplementary material that are now found in Figure 6—figure supplement 1.

2) The fact that the RGC-depleted retinae upon Atoh7 depletion keep their curvature and shape at both apical and basal sides (Figure 6A-B), strongly argues that a hypothetical basal-apical contractile force is not perturbed after RGC-loss and thereby not dependent on RGC layer formation.

3) In the *atoh7* depleted retinae, the most basal layer is filled with ACs and HCs (Weber et al., 2014, Kay et al. 2004, Kay et al. 2001)*.* Hence, this most basal layer is occupied by other retinal cell-types and does not remain empty, explaining the maintenance of overall tissue architectures as also shown in Randlett et al., 2011.

This modified model is, in my opinion, already at least partly supported by the authors' RGC-ablation experiment, and it may also well harmonize with "the path of least resistance" model suggested for amoeboid migration (Renkawitz et al. 2019).

This is an excellent point. We agree with the reviewer’s idea and we also think that our RGC-depleted as well as LMN-A overexpression data suggest that amoeboid-like migratory HCs likely follow "the path of least resistance" as was previously shown by Sixt lab for dendritic cells in vitro (Renkawitz et al. 2019).

This point was added in the discussion part of the manuscript and follow up studies in the lab will investigate this point further.

Additional experiments, for example, to loosen the basal lamina, removing laminin by morpholino (Randlett et al. Neuron 2011) or by antibody injection into the basal lens cavity just before MHC migration period or injecting collagenase (done in chick, Halfter J Comp Neurol 397, 89, 1998), could be considered if feasible.

We have performed this experiment in our study from 2016, Icha et al. (doi: 10.1083/*jcb*.201604095), where we depleted laminin and evaluated its impact on RGC migration. We there noted:

“we imaged the laminin morphants between 36 hpf and 84 hpf in the same embryo every 12 hours using LSFM. This revealed that when RGCs arose, the whole retinal tissue detached from the basal side and ruptured (Figure S3C and S3D). As the laminin morphant data were hard to interpret, we concentrated on the interference with intracellular components important for BP attachment.”

For this reason, we believe that the full depletion of RGCs was the more reliable and reproducible experiment to remove RGCs and look for effect on committed HCs precursors and HCs migration.

- Would additional experiments be essential to support the claims of the paper? Request additional experiments only where necessary for the paper as it is, and do not ask authors to open new lines of experimentation.Are there any correlations between the distribution of dividing (rounding up) MHCs (HCpr) and/or bipolar cells (BPpr), which are more frequent in the apical half of INL than in the basal half (2019 Development, by the authors), and Brillouin signals? I ask this because amoeboid migrating MHCs interestingly change their ways upon contacting with dividing cells (Figure 3), as if a hurricane or typhoon is repelled by a high pressure. From the same point of view, how Brillouin signals, especially elasticity, change when RGCs are depleted and when overproliferation to increase tissue volume/pressure?

Our results show that migrating HCs also undergo direction changes and morphological deformations in the absence of mitotic cells in their immediate vicinity (e.g. when there are squeezing out of the AC layer or in later developmental stages wherein mitosis is minimal). To better clarify this point that such circumventions does not only occur upon bumping into a neighboring mitotic cell, we added additional examples and experiments which can now be found in the text and Figure 3—figure supplement 2.

How elasticity changes in RGC-depleted retinae or under over proliferation conditions, the exact mechanical interplay and forces between HCs and their environment in the above-mentioned perturbed scenarios and even in physiologically unperturbed conditions will be further investigated in an independent study that is already started in the laboratory.

Minor comments:- Specific experimental issues that are easily addressable.In their previous paper (Biophys J 2018), the authors showed both the Brillouin shift (GHz) and the longitudinal modulus (GPa) (as well as the viscosity [mPa]) side by side, and then stated that Brillouin signals constitute viscoelastic properties. In the present study, Figure 2D-E show Brillouin shift maps (GHz) only, with no longitudinal modulus and viscosity and with no careful/kind explanation for compatibility between them. So, I am afraid that simply telling that Brillouin shift maps (GHz) indicate "compressibility", as in the text of the current version, is inappropriate.In the publication mentioned (Schlüßler et al., Biophys J 2018) the authors conclude that the refractive index has “very little, if not negligible, contribution” to the longitudinal modulus and show a very high correlation (correlation coefficient of 0.9968) between the Brillouin shift and the longitudinal modulus for zebrafish tissue. Hence, the Brillouin shift presented here directly resembles the longitudinal modulus and, hence, the compressibility of the sample.

In the publication mentioned (Schlüßler et al., Biophys J 2018) the authors conclude that the refractive index has “very little, if not negligible, contribution” to the longitudinal modulus and show a very high correlation (correlation coefficient of 0.9968) between the Brillouin shift and the longitudinal modulus for zebrafish tissue. Hence, the Brillouin shift presented here directly resembles the longitudinal modulus and, hence, the compressibility of the sample.

We adjusted the respective section to clarify this point.

– Are prior studies referenced appropriately?Mostly yes.Hinds and Hinds (1979, J Comp Neurol 187) should be cited for their 3D EM reconstruction of the mouse retinal HCs including an MHC.

We added this reference.

Santos et al. (2020, Cell Rep 32) should also be cited for amoeboid growth of neuronal axons. Yamada and Sixt (2019, Nat Rev Mol Cell Biol) and Lammermann and Sixt (2009, Curr Opin Cell Biol 21) are excellent review papers, helpful to readers.

We cited these reviews.

– Do you have suggestions that would help the authors improve the presentation of their data and conclusions? Overall well presented.Figure 1B is vague about what the question is.

As noted above, we now amended schematics and legends along the manuscript.

Figure 2DE are in question (longitudinal modulus and viscosity could be added, and hopefully simultaneous imaging of M-phase cells and Brillouin microscopy if technically possible, in Figure 3).

See our response to the first minor comment:

“In a previous publication (Schlüßler et al., Biophys J 2018) the authors conclude that the refractive index has “very little, if not negligible, contribution” to the longitudinal modulus and show a very high correlation (correlation coefficient of 0.9968) between the Brillouin shift and the longitudinal modulus for zebrafish tissue. Hence, the Brillouin shift presented here directly resembles the longitudinal modulus and, hence, the compressibility of the sample. We will adjust the respective section to clarify this point.”

Further, simultaneous confocal fluorescence imaging and Brillouin microscopy is not possible as different devices for the two modalities are used. Brillouin measurements of cells in a specific cell cycle phase are impossible, since a Brillouin measurement takes ranges of up to one hour or more for a decently resolved map while M-phase only takes 20minutes. Also, at the stages at which Brillouin measurements are performed, most cells are already neurons and will not undergo actively cycle any longer.

Reviewer #3 (Evidence, reproducibility and clarity (Required)):SummaryUsing in vivo microscopy in the developing zebrafish retina, horizontal cells (HCs) migrating along intercellular clefts between neuroepithelial cells were monitored and the migration type and guiding tissue structures examined. Extracellular cues guiding HC migration were identified, including the lack of clear ECM cues, but narrow intercellular clefts, along which the cells move. Additional analyses show vigorous nuclear deformation, a dynamic cortical actin cytoskeleton generating pointed protrusions while lacking defined focal adhesions, and a well-developed uropod at the cell rear. Ectopic overexpression of nuclear lamin A caused a migration delay and compromised path finding. The authors conclude that HCs utilize physics-based amoeboid-like protrusiveamoeboid movement along a multicellular scaffold. Lastly, they show that the inner plexiform layer precludes HC movement and causes change of directionality and conclude, and interpret this as barrier mechanism.Major comments1) The absence of structural tissue remodeling was tested morphologically, by visualizing cell-cell boundaries with light microscopic resolution. However, proteases expressed by HCs may still degrade cadherin junctions at a molecular level and facilitate intercellular movement, by opening a deformable trail. This should be formally ruled out, using e.g., pharmacological protease inhibition.

We thank the reviewer for raising this issue.

While we do not exclude a role of proteases-dependent remodeling to open a deformable trail, we consider it unlikely given the nature of the tissue; (1) lack of prominent ECM components, (2) the dense packing of retinal cells and their nuclei, (3) size of nuclei (Figure 2—figure supplement 1C) and the fact that there is minimal cytoplasm between the nucleus and cell membrane (Figure 2—figure supplement 1C-C’). Hence, degrading cadherin junctions and cell membranes is unlikely to accommodate HC’s passage through the densely-packed retinal nuclei. In addition, in *atoh7* morphant conditions, HCs remain trapped below the IPL without severing the fibrillar network of axonal and dendritic processes to create a deformable path through the IPL. Lastly, according to our experience, protease inhibition treatment by protease inhibition cocktail is unspecific to address this question as it leads to overall tissue disintegration in the animal. As we cannot fully exclude the possibility of protease-based tissue remodeling at this stage, we have toned down our statement and adjusted the text in the respective sections in results and discussion.

2) The idea that the inner plexiform layer acts as a barrier is incompletely developed, because mechanical and chemical barrier functions were not discriminated. Can a purely mechanical barrier suffice to cause such stringent reorientation of HCs, followed by migration into the opposite direction?

In principle, “Chemical repulsive barriers” could mediate repolarization of the migrating cells and steering them away from the original direction during contact-repulsion. However, we do not observe that HCs reorient or move in opposite direction upon hitting the IPL but rather remain trapped underneath it, while moving in all directions. This suggests that the IPL does not act as a “chemical barrier”. Given that the IPL shows enrichment of membranes of the neuronal termini (Figure 2—figure supplement 1C-C’’) and that therefore space is limited, we believe it is likely that HCs cannot overcome this spatial constraint. We now amended the text and added more data on this point in Figure 6—figure supplement 1.

We further added in the discussion that we cannot exclude that some chemical factors might be also involved.

3) Candidate molecular mechanisms of cell repulsion have not been ruled out. The relative contribution of mechanical versus repulsive signals should be worked out using direct interventions, e.g., by interfering with repulsive signals expressed by retinal ganglion cells (Ephrin/Eph, etc.).

This is an excellent point brough by reviewer and we are currently generating a transcriptomics dataset for follow up investigations that can address what attractive and/or repulsive chemical cues trigger different phases of HC migration and layer formation.

We believe that due to the plethora of possible candidates such an approach would be beyond the scope of this study. As stated above, we added in the discussion the fact that a role for additional chemical factors, in particular from the IPL, cannot be excluded.

Regarding the reviewer’s comment that “repulsive signals expressed by retinal ganglion cells” could contribute to HCs migration behavior, we do not think that repulsion by retinal ganglion cells contributes to HC migration and layer formation for the following reasons:

1) Data from our previous study (Amini et al., 2019) shows that before formation of the IPL, HCs and their precursors can migrate to more basal position and even undergo mitosis within the retinal ganglion cell layer before returning apically. This argues against a role for chemical repulsive cues from the retinal ganglion cells per se in steering HCs away from the ganglion cell layer.

2) As for why depth of HC migration is significantly shallow in controls in comparison to *atoh7* morphants, our data here and in our previous study (Amini et al., 2019) shows a direct role between IPL formation and depth of migration but no contribution from the retinal ganglion cells.

3) In *atoh7* morphants, the most basal retinal layer is filled with ACs and HCs as retinal ganglion cells are absent in this condition. Hence, we expect that without retinal ganglion cells their potential repulsive signals (Ephrin/Eph, etc.) would be also absent. Nevertheless, even in this condition before IPL formation, majority of HCs successfully reach their destination despite the fact that they need to travel from more basal positions (adjacent to basement membrane). This suggests that retinal ganglion cells and their potential repulsive cues do not contribute to HC migration and layer formation.

Minor comments"The developing zebrafish retina is a densely-packed environment (Matejcic, Salbreux and Norden 2018) which undergoes structural changes in space and time during neuronal lamination." – Additional information should be provided detailing the interstitial organization and potential guidance paths. In addition, besides basement membrane proteins, fibrillar collagen should be examined, to rule out interstitial ECM cues.

We performed collagen stainings that are now shown in Figure 2—figure supplement 1A. Similar to the other ECM components, while we observed anti-Collagen IV in the basal lamina, we did not detect collagen in the INL in which HCs migrate.

Using Brillouin shift analysis, a viscoelasticity gradient of the retinal tissue was probed. The authors conclude that based on "(…) the characteristic features of each retinal layer (…), Brillouin shift values are influenced by nuclear occupation (…)". However, the correlation of Brillouin shift values and nuclear positions are not sufficiently correlated from the images provided in Figure 2C, D. It would be helpful to show the nuclear positioning, using fixed samples in association with higher-resolved zooms of the Brillouin shift measurements.

This has actually been addressed in a previous study by Sanchez Iranzo et al., 2020

DOI:0.1016/j.dib.2020.105427, where the authors indeed show that nuclear occupation and Brillouin shift values correlate similarly to what is presented here in a 52hpf old zebrafish retina. We clarified this point and added this citation to the text.

"We showed that migrating HCs navigate within a crowded environment with neither cellular nor ECM scaffolding structures" – —figure supplement Figure 2 shows that HC move along cell-cell boundaries. Are these not cellular scaffolding structures?

We clarified this in the results and Discussion sections.